# Feature Matching Intervention: Leveraging Observational Data for Causal Representation Learning

## Abstract

A major challenge in causal inference from observational data is the absence of perfect interventions, making it difficult to distinguish causal features from spurious ones. We propose an innovative approach, Feature Matching Intervention (FMI), which uses a matching procedure to mimic perfect interventions. We define causal latent graphs, extending structural causal models to latent feature space, providing a framework that connects FMI with causal graph learning. Our feature matching procedure emulates perfect interventions within these causal latent graphs. Theoretical results demonstrate that FMI exhibits strong out-of-distribution (OOD) generalizability. Experiments further highlight FMI's superior performance in effectively identifying causal features solely from observational data.

## 1 Introduction

Causal representation learning (Schölkopf et al., 2021) aims to uncover causal features from observations of high-dimensional data, and is emerging as a prominent field at the intersection of deep learning and causal inference. Unlike traditional causal effect of a specific treatment variable, causal representation learning does not treat any observed variable as a potential causal parent. Instead, it focuses on transforming the observational space into a low-dimensional space to identify causal parents.

However, despite its promise, recent years have witnessed notable shortcomings in effectively capturing genuine causal features, particularly evident in tasks such as image classification. Numerous experiments over the past decade (Geirhos et al., 2020; Pezeshki et al., 2021; Beery et al., 2018; Nagarajan et al., 2020) have highlighted the failure of models to discern essential features, resulting in a phenomenon where models optimized on training data exhibit catastrophic performance when tested on unseen environments. This failure stems from the reliance of models on spurious features within the data, such as background color in images, rather than the genuine features essential for accurate classification, such as the inherent properties of objects depicted in the images. Consequently, models are susceptible to errors, particularly when faced with adversarial examples.

The phenomenon described above is commonly known as out-of-distribution (OOD), with efforts to mitigate it termed as out-of-distribution generalization or domain generalization. To tackle this challenge, numerous approaches have been proposed. Among the most significant concepts is the invariance principle from causality (Peters et al., 2016; Pearl, 1995), which forms the basis of invariant risk minimization (IRM) (Arjovsky et al., 2019). IRM was the first to operationalize this principle, aiming to identify invariant features through data from multiple environments. The invariance principle dictates the optimal predictor based on invariant features, ensuring minimal risk across any given environment (Rojas-Carulla et al., 2018; Koyama & Yamaguchi, 2020; Ahuja et al., 2020). Additionally, several works have extended IRM by imposing extra constraints on the invariance principle (e.g., Krueger et al. (2021); Ahuja et al. (2021); Chevalley et al. (2022)).

Despite the promise of IRM and the invariance principle under certain assumptions, subsequent research has revealed their limitations (Rosenfeld et al., 2020). Moreover, invariance does not necessarily imply causality universally. All state-of-the-art methods struggle to distinguish between spurious and true features without a perfect intervention. In other words, the absence of perfect

interventions is the primary challenge for all current approaches, as it makes it difficult to reliably distinguish between spurious and true features.

In this paper, we propose a very simple alternative approach to learning causal representations through covariate matching. This approach attempts to emulate perfect interventions, which is known to be a difficult problem. Our method eliminates the need for multiple environmental datasets and does not rely on the use of an invariance algorithm. We make only the verifiable assumption that spurious features are present in the training data, a scenario commonly encountered in practice. While covariate matching is a traditional method in the statistics literature, it has been less explored in causal representation learning. Leveraging the causal latent graph introduced later in the paper, our matching algorithm offers an explicit interpretation of emulating perfect intervention on the spurious feature. We demonstrate the effectiveness of our approach through unit tests (Aubin et al., 2021) and experiments on image datasets. Our main contributions can be summarized as follows:

**Contributions.** (1) We propose an innovative and straightforward algorithm – FMI based on covariate matching in the presence of spurious feature in the training data. By emulating perfect intervention on the spurious feature, we are able to learn underlying causal feature (2) We propose an approach to test the assumption of spurious feature being learned in the training environment. (3) We validate our matching algorithm using a causal latent graph. (4) Our experiments on unit tests, Colored MNIST, and *WaterBirds* datasets demonstrate the superior performance of our algorithm compared to state-of-the-art methods.

## 2 PRELIMINARIES

Let $\{(x_i, y_i)\}_{i=1}^n$ be our training data where $x_i \in \mathcal{X}$ and $y_i \in \mathcal{Y}$. In the theory part of this paper, we consider the case $\mathcal{X} = \mathbb{R}^d$ and $\mathcal{Y} = \{0, 1\}$. Similar results still hold for other spaces (e.g., when $\mathcal{Y}$ contains more than 2 values). Let $\ell(\cdot, \cdot) : \mathcal{X} \times \mathcal{Y} \to \mathbb{R}$ be our loss function (e.g., cross-entropy or 0-1 loss), and $\mathcal{R}(\cdot) : \mathcal{M} \to \mathbb{R}$ be our risk function, where $\mathcal{M}$ is the model space. Suppose each $(x_i, y_i)$ follows the distribution $P_{\text{tr}}(X, Y)$. The major problem in domain generalization is that the test data distribution $P_{\text{te}}(X, Y)$ differs from the training distribution $P_{\text{tr}}(X, Y)$, making it challenging and crucial to identify causal features. To better describe the shift of the distribution, we can consider a set $\mathcal{E}$, namely the environment set. The joint distribution of $(X, Y)$ can be indexed by this set $\mathcal{E}$, i.e., for $e_1 \neq e_2 \in \mathcal{E}$, $P^{e_1}(X, Y) \neq P^{e_2}(X, Y)$. In a similar fashion, we denote the risk of a model $f \in \mathcal{M}$ on a certain environment $e$ by $\mathcal{R}^e(f)$. Note that in practice, we rarely observe all environments, or even multiple environments. In this paper, the training data corresponds to only one environment in $\mathcal{E}$.

To tackle this problem of domain generalization, we aim to learn causal feature from the training data. To this end, we follow common techniques from the traditional causal inference literature (Peters et al., 2017; Pearl, 2009) to model the data generating process of our model using a causal latent graph:

**Definition 1** (Causal latent graph). *For any environment $e \in \mathcal{E}$, the causal latent graph of a given dataset is defined as a directed acyclic graph $\mathcal{G}^e = (V^e, E^e)$ with $V^e = (Z_{\text{spu}}^e, Z_{\text{true}}^e, Y^e)$ such that $\text{pa}(Y^e) = \{Z_{\text{true}}^e\}$ and $Z_{\text{spu}}^e \notin \text{an}(Y^e)$, where $\text{pa}(\cdot), \text{an}(\cdot)$ represent the parent set and the ancestor set of a node, respectively.*

**Remark on Definition 1.** By Reichenbach's common cause principle (Reichenbach & Morrison, 1956), as long as $Z_{\text{spu}}^e \not\perp Y^e$ in environment $e$, then either (1) $Z_{\text{spu}}^e \in \text{an}(Y^e)$, or (2) $Y^e \in \text{an}(Z_{\text{spu}}^e)$, or (3) there is a common ancestor of $Z_{\text{spu}}^e$ and $Y^e$. Note that since we assume there is no hidden variable in the latent graph and $Z_{\text{spu}}^e \notin \text{an}(Y^e)$, we rule out all DAGs except for the two cases shown in Figure 1. In fact, these two types of causal graphs correspond to the two cases specified in Ahuja et al. (2021) (See Appendix C), namely fully informative invariant features (FIIF) and partially informative invariant features (PIIF). We express them in a unified framework in our setting.

The observed covariate $X^e$ is a mapping from latent space through a mixing function $g$, i.e., $X^e = g(Z_{\text{spu}}^e, Z_{\text{true}}^e)$. To learn the feature from a training environment, denoted as $e_0$, the state-of-the-art

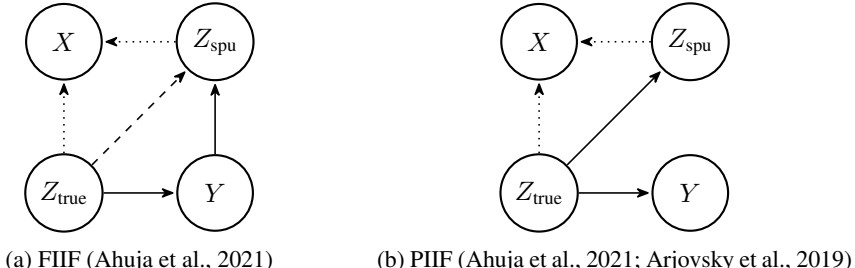

(a) FIIF (Ahuja et al., 2021)    (b) PIIF (Ahuja et al., 2021; Arjovsky et al., 2019)

Figure 1: Possible latent DAGs: (a) corresponds to the FIIF case and (b) corresponds to the PIIF case.

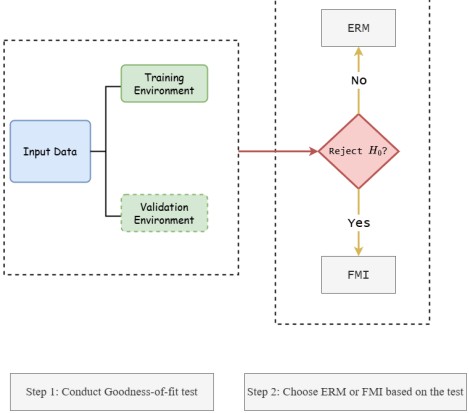

Step 1: Conduct Goodness-of-fit test    Step 2: Choose ERM or FMI based on the test

Figure 2: The workflow of FMI: Given training data, we can conduct a hypothesis test based on another validation environment. If we get a rejection, we apply FMI to learn the true feature. Otherwise, we can use the feature learned by ERM.

model would solve the following optimization problem so that it is able to find the Bayes classifer:

$$f^{e_0} = \arg\min_{f \in \mathcal{M}} \mathcal{R}^{e_0}(f). \tag{1}$$

However, due to the shift in distribution, $f^{e_0}$ does not necessarily minimize the risk on a new environment (testing environment). In such scenarios, it is essential to train a model that performs well across all possible environments. Equivalently, the ideal model we are seeking should solve the following minimax problem:

$$f^* = \arg\min_{f \in \mathcal{M}} \max_{e \in \mathcal{E}} \mathcal{R}^e(f). \tag{2}$$

Ideally, interventions are required for causal representation learning. Even if we aim to decide directly in the high-dimensional space whether $X$ is the cause of $Y$, intervention, especially perfect intervention is needed. We will show in Section 4 that we can emulate perfect intervention with training data, thereby to achieve causal representation learning.

In this paper, we consider the model space $\mathcal{M} = \{f \circ \phi | \phi : \mathcal{X} \to \mathcal{H}, f : \mathcal{H} \to \mathcal{Y}\}$, where $\mathcal{H}$ is the space of feature. It is worth noting that for a given model $f \circ \phi$ and any invertible transformation $\psi$, $f \circ \phi = (f \circ \psi^{-1}) \circ (\psi \circ \phi)$. Thus, identifiability becomes an issue here. However, since our goal is to learn $f \circ \phi$, this concern is not relevant. Henceforth, we assume $\phi$ and $f$ are two neural networks with fixed architecture. We assume $\phi$ is parameterized by $\theta_\phi$ and $f$ is parameterized by $\theta_f$. We refer to $\phi(\cdot; \theta_\phi)$ as the featurizer and $f(\cdot; \theta_f)$ as the classifier. For simplicity, we denote the entire model parameterized by $(\theta_f, \theta_\phi)$ as $f \circ \phi(\theta_f, \theta_\phi)$. Our goal is to find a model that solves problem (2) under certain conditions and the corresponding feature $\phi$ will then define a causal representation feature.

## 3 RELATED WORK

We summarize below some relevant works from previous literature.

**Invariance-based Domain Generalization** Numerous works in the literature have studied domain generalization, a problem closely related to causal inference. Many of these works aim to discover the invariant predictor, as demonstrated by studies such as Arjovsky et al. (2019); Ahuja et al. (2021); Chevalley et al. (2022); Yuan et al. (2023). The concept of invariance originates from causal inference as a necessary condition of causal variables (Peters et al., 2016; Bühlmann, 2020). The fundamental idea behind these methods is the utilization of data from multiple environments (domains), either through deliberate design or collection. For instance, Arjovsky et al. (2019) necessitates the environments to be in linear general position. However, these approaches exhibit a voracious appetite for environments, and the invariance principle reduces the objective function to that of the standard empirical risk minimization (ERM) (Vapnik, 1991) when only one environment is available.

**Causal Representation Learning** Ahuja et al. (2023) provided identifiability results of latent causal factors using interventional data. Buchholz et al. (2024) demonstrated identifiability results under the assumption of a linear latent graphical model. Additionally, Jiang & Aragam (2024) established conditions under which latent causal graphs are nonparametrically identifiable and can be reconstructed from unknown interventions in the latent space. These results motivated us to emulate intervention with observational data and therefore achieve causal representation learning.

The most relevant works to ours are MatchDG (Mahajan et al., 2021) and Chevalley et al. (2022). In MatchDG, the authors employ a matching function to pair corresponding objects across domains and seek features with zero distance on these matched objects while minimizing the loss on the training data. However, there are at least two distinctions between our approach and MatchDG: (1) We do not require data from multiple domains; (2) Our approach matches training data based on their classification results of the spurious features extracted by the subnetwork, eliminating the need for any matching function.

In Chevalley et al. (2022), the authors randomly partition each minibatch into two groups and penalize the distance between the latent features learned from each group. In contrast, our method does not necessitate random separation; Instead, it emulates perfect intervention, a guarantee not provided in the previous work.

## 4 SINGLE TRAINING ENVIRONMENT: FEATURE MATCHING INTERVENTION (FMI)

In this section, we introduce a novel algorithm called Feature Matching Intervention (FMI) for causal representation learning that solves problem (2). The idea behind FMI is that, since ERM builds the model with the spurious feature, why not exploit this additional information and use the result of ERM to control this spurious feature through a matching procedure? Matching has been a well-known method in the causal inference literature for estimating treatment effects from observational data (Stuart, 2010). Leveraging this concept, we aim to develop a method for matching the spurious feature in the training environment and then training the model after this matching process. With this matching procedure, we can emulate perfect intervention on the spurious feature. Example 1 demostrates this matching idea.

**Example 1.** *Consider the task of classifying images containing the digits* 0 *and* 1*. Suppose a large proportion of images with the digit* 0 *happen to be colored red, and a large proportion of images with the digit* 1 *happen to be colored green. Then the Bayes classifier in the training environment would be based on color. Figure 3 visualizes the classification result in the training environment as well as the matching process. Clearly, after the matching process, the label and the color become uncorrelated. Therefore, the matching corresponds to an intervention on the spurious feature (color).*

To formally define our matching approach, suppose we have training data $(X_{\text{tr}}, Y_{\text{tr}}) \sim P^{e_0}(X, Y)$. Let $(\theta_f^{e_0}, \theta_\phi^{e_0})$ be a solution to problem (1) and $(\theta_f^*, \theta_\phi^*)$ to problem (2). That is,

$$(\theta_f^{e_0}, \theta_\phi^{e_0}) = \underset{(\theta_f, \theta_\phi)}{\arg\min}\, \mathcal{R}^{e_0}(f \circ \phi(\theta_f, \theta_\phi)), \tag{3}$$

and

$$(\theta_f^*, \theta_\phi^*) = \underset{(\theta_f, \theta_\phi)}{\arg\min}\, \underset{e \in \mathcal{E}}{\max}\, \mathcal{R}^e(f \circ \phi(\theta_f, \theta_\phi)). \tag{4}$$

We can similarly define $(\theta_f^e, \theta_\phi^e)$ to be the Bayes classifier in environment $e$, i.e.,

$$(\theta_f^e, \theta_\phi^e) = \underset{(\theta_f, \theta_\phi)}{\arg\min}\, \mathcal{R}^e(f \circ \phi(\theta_f, \theta_\phi)).$$

For simplicity, assume there are 2 classes (the formula for a general case with more classes can be similarly derived). Now, we can define a new environment $e_m$ by subsampling from the training data. More specifically, let $\hat{f}$ be the ERM solution in the training data, i.e., the predicted label. We subsample from the training data so that the spurious feature is balanced in the matching environment $e_m$. Our matched samples satisfy the following:

$$
\begin{aligned}
P^{e_m}(Y = 0|\hat{f} = 0) = \frac{1}{2}, \quad & P^{e_m}(Y = 1|\hat{f} = 0) = \frac{1}{2} \\
P^{e_m}(Y = 0|\hat{f} = 1) = \frac{1}{2}, \quad & P^{e_m}(Y = 1|\hat{f} = 1) = \frac{1}{2}.
\end{aligned}
\tag{5}
$$

**Remark on Equation (5).** This is a sample version of conditional independence. To make $Y$ independent of the learned feature in the training environment, ideally we need access to the population version of $\hat{f}$. Nevertheless, in this new environment $e_m$, $Y$ and $\hat{f}$ are independent because it holds $P^{e_m}(Y = 0) = P^{e_m}(Y = 1) = P^{e_m}(Y = i|\hat{f} = j), i, j \in \{1, 2\}$. Also, this distribution of the subsample is equivalent to an interventional distribution.

**Proposed approach** FMI solves the following optimization problem:

$$(\theta_f^{\text{FMI}}, \theta_\phi^{\text{FMI}}) = \underset{(\theta_f, \theta_\phi)}{\arg\min}\, \mathcal{R}^{e_m}(f \circ \phi(\theta_f, \theta_\phi)), \tag{FMI}$$

where the risk is with respect to the distribution $P^{e_m}$ we defined before.

The rationale behind this formulation is that if we know the ERM classifier will converge to the Bayes classifier in the training environment, then the classification result of the learned ERM classifier should be based purely on the spurious feature. Therefore, by subsampling from the training data as in Formula (5), the true label and the predicted label (which depends only on the spurious feature) in the subsample are independent. This property is also satisfied when there is perfect intervention (See Appendix D) on the spurious feature. In fact, this matching approach can be considered a special kind of intervention and by doing so, we manage to emulate perfect intervention on the spurious feature.

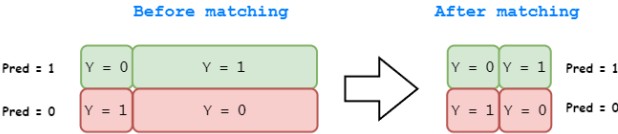

Figure 3: Illustration of the matching approach: The Bayes classifier classifies green images as 1 and red images as 0. Although it achieves a risk smaller than that of the true feature (digit shape), it performs poorly in other environments. FMI subsamples according to another distribution from the original training environment and therefore balances the spurious feature (color). In this new distribution, we should expect the Bayes classifier to be based on the true feature.

---

**Algorithm 1** Feature Matching Intervention (FMI)

---

1: Let $n > 0$ be the number of samples drawn at each step;
2: Let $f_1, f_2$ be two Neural Networks;
3: **begin**
4: Draw a batch $b_i$ of $n$ samples;
5: **if** train $f_1$ **then**
6:    update the parameters of $f_1$ using this $b_i$; {Use a variable to control whether train or not}
7: **end if**
8: subsample $b_s$ from $b_i$ following Formula (5);
9: update the parameters of $f_2$ using $b_s$;
10: **return** loss;
11: **end**

---

Previous research has emphasized the crucial role of perfect intervention in identifying latent causal representations (Ahuja et al., 2023; Buchholz et al., 2024; Jiang & Aragam, 2024). In Buchholz et al. (2024), the authors demonstrated the significant impact of the number of perfect interventions on latent variables to the identifiability of latent causal graphs. The proposed matching approach aims to emulate a form of perfect intervention on the latent spurious variable, overcoming a limitation of environment-based algorithms, as they often impose strong assumptions on the variability of the environment (e.g., linear general position in Arjovsky et al. (2019)).

The algorithm of FMI is shown in Algorithm 1. In practice, we can train two neural networks at the same time (one of them learns the spurious feature and will be used in (5)) , as we show in Appendix A. With the same number of steps, training them together can make FMI perform better.

## 5 THEORETICAL GUARANTEE OF FMI

### 5.1 MAIN RESULT

It turns out that, under appropriate assumptions we are able to learn $\phi(X; \theta_\phi^*) = Z_{\text{true}}$ through (FMI).

In many scenarios, the feature learned in the training environment cannot perform equally well on new environments. In those cases, it is highly plausible that the feature learned by solving (1) is the spurious feature. Below, we introduce an assumption that accounts for this issue, which essentially serves as an identifiability statement.

**Assumption 1.** *Given training environment $e_0$, the model $f^{e_0}$ learned by solving (1) is based on $Z_{\text{spu}}^{e_0}$.*

Although this is a requirement on the identifiability of the spurious feature, we will show in Section 5.2 that with some extra information about the environment, we are able to test whether this assumption holds or not.

Below, we make two assumptions about the structural equation in the latent causal graph and the environment set:

**Assumption 2.** *In each $e \in \mathcal{E}$,*

$$Y^e \leftarrow \mathbb{I}(w_{\text{true}} \cdot Z_{\text{true}}^e) \oplus N^e, \quad N^e \sim \text{Bernoulli}(q), q < \frac{1}{2}, \quad N^e \perp\!\!\!\perp Z_{\text{true}}^e,$$

$$X^e \leftarrow S(Z_{\text{spu}}^e, Z_{\text{true}}^e),$$

*where $w_{\text{true}}$ with $\|w_{\text{true}}\| = 1$ is the labelling hyperplane, $Z_{\text{true}}^e \in \mathbb{R}^m$, $Z_{\text{spu}}^e \in \mathbb{R}^o$, $N^e$ is binary noise with identical distribution across environments, $\oplus$ is the XOR operator, $S$ is invertible.*

Now, Given the definition of the latent causal graph and structural equations, we assume that any environment $e \in \mathcal{E}$ comes from a specific set of intervention in the graph:

**Assumption 3.** *The environment set $\mathcal{E}$ contains all interventions on $Z_{\text{spu}}$, $Z_{\text{true}}$. For each environment $e \in \mathcal{E}$, the distribution $P^e(X, Y)$ corresponds to the interventional distribution of $\mathcal{G}^e$ (See Appendix D).*

**Remark on Assumption 2.** It is worth noting that neither $Z_{\text{spu}}$ nor $Z_{\text{true}}$ is known to us initially. Additionally, since the solutions to (1) and (2) are not unique, we fix the classifier $f$ to be the indicator function (considering 2-class classification) and put everything else in the feature from now on. Also, we only consider linear classifiers in our theory.

**Remark on Assumption 3.** This assumption is similar to the one made in Ahuja et al. (2021). However, our assumption encompasses both the fully informative invariant features (FIIF) and partially informative invariant features (PIIF) cases, making it more general. Another crucial aspect of this assumption is that $Z_{\text{spu}}$ in the causal latent graph is well-defined: it is the feature learned in the training environment. This information proves valuable as it provides an opportunity to 'correct' the mistake the classifier made in the training environment, thereby enhancing generalizability. Furthermore, we assume that the causal latent graph we defined contains all information about the joint distribution of the observations $X$ through an invertible link function $S$.

We will show in Appendix B that, any classifier that achieves the optimal risk in a specific environment has the same risk and the same decision boundary. However, as readers will notice, it is the dependence of the spurious feature and the true feature in that environment that allows the spurious feature to achieve the optimal risk.

The last assumption we make is about the support of the features.

**Assumption 4.** *The support of $Z_{\text{true}}$ and $Z_{\text{spu}}$ both contain some circle centered at zero and do not change across environments.*

**Remark on Assumption 4.** This assumption is a regularity condition for the features. Note that the zero-centering constraint can be relaxed, as it only requires an affine transformation.

Now, we are ready for our main result.

**Theorem 1.** *Under Assumptions 1-4, any solution to (FMI) achieves the minimax risk as in Formula (2). Therefore, FMI offers OOD generalization.*

**Proof sketch.** First, we prove that in any environment, the optimal solution in that environment achieves an error of $q$ — the noise level — and also has the same decision boundary as $\mathbb{I}(w_{\text{true}} \cdot Z_{\text{true}})$. Then we show that the optimal solution to (FMI) only uses $Z_{\text{true}}$ in the decision boundary.

Theorem 1 serves as the theoretical guarantee of FMI — it means the solution to (FMI) solves the OOD generalization problem with respect to the entire set of environments.

### 5.2 ASSEMENT OF THE FEATURE LEARNED IN THE TRAINING ENVIRONMENT

In Assumption 1, we assumed that the feature learned in the training environment is spurious. This assumption might not hold in practice: the feature learned in the training environment could be the true feature, or even the mixture of true feature and spurious feature. In order to verify our assumption, we propose a method that facilitates a validation environment. Below, we give the definition of a validation environment:

**Definition 2** (Validation environment for feature). *An environment $e \in \mathcal{E}$ is a validation environment for feature $Z$ if the conditional distribution $Y^e | Z^e$ is different from $Y^{e_0} | Z^{e_0}$.*

Clearly, if we can find this validation environment, then we have enough reason to reject the feature learned in the training environment. In fact, under Assumptions 1 and 3, we have the following result:

**Proposition 1.** *Under Assumption 1 and Assumption 3, there exist validation environments for the feature learned in the training environment.*

**One line proof.** Let $e \in \mathcal{E}$ defined by an intervention on $Z_{\text{spu}}$ such that $Y^e \perp\!\!\!\perp Z^e_{\text{spu}}$, then $e$ is a validation environment for $Z_{\text{spu}}$.

In fact, there exist infinite number of environments in this case: since $Y^e$ is discrete, we can find an intervention $e \in \mathcal{E}$ such that the distribution of $Y^e | Z^e_{\text{spu}}$ differs from $Y^{e_0} | Z^{e_0}_{\text{spu}}$ arbitrarily.

In practice, given the access to an environment other than $e_0$, we can validate whether there is evidence to believe the assumptions we made. Specifically, with $Y$ being discrete, we can apply a goodness-of-fit test on $Y^e|\phi(X^e; \theta^{e_0}_\phi)$ and $Y^{e_0}|\phi(X^{e_0}; \theta^{e_0}_\phi)$. To conduct this test, we only need access to a sample from the new environment $e$. The details of the goodness-of-fit test can be found in Appendix E. If the test result is significant, Assumption 1 holds, and FMI is the better choice for learning causal features. However, if the test result is insignificant, it suggests that the features identified by an existing algorithm, such as ERM, are the causal ones, and there is no need to implement FMI for improvement.

In Section 6, we conduct experiments to show the effectiveness of this test.

# 6 EXPERIMENTS

The details of all our experiments can be found in Appendix A.

## 6.1 SYNTHETIC EXPERIMENT

We first conduct an experiment on a synthetic dataset. This dataset is from Example 2/2S of unit tests proposed in Aubin et al. (2021) and originated from the famous cow-camel example (Beery et al., 2018). The data generating process corresponds to Figure 1(a). We compare FMI with four approaches: ERM (Vapnik, 1991), ANDMask (Parascandolo et al., 2020), IGA (Koyama & Yamaguchi, 2020), and IRM (Arjovsky et al., 2019).

**Conclusion** Although all methods except FMI and ERM benefit from multiple environments, the classification errors on the testing data for all algorithms **except FMI** are approximately 50% as shown in Table 1, indicating that none of them successfully captures the causal feature from the training data. Howver, FMI can capture the true feature from the training data and therefore achieves zero testing error.

Table 1: Performance Comparison of Various Methods on Example 2/2S. The lowest number in each row, representing lowest classification error on testing data, is boldfaced. The oracle is obtained by running ERM on testing data.

| Method | ANDMask | ERM | FMI | IGA | IRM | Oracle |
|--------|---------|-----|-----|-----|-----|--------|
| Example2.E0 | $0.43 \pm 0.00$ | $0.39 \pm 0.01$ | $\mathbf{0.00 \pm 0.00}$ | $0.43 \pm 0.01$ | $0.43 \pm 0.01$ | $0.00 \pm 0.00$ |
| Example2.E1 | $0.49 \pm 0.01$ | $0.45 \pm 0.02$ | $\mathbf{0.00 \pm 0.00}$ | $0.50 \pm 0.01$ | $0.50 \pm 0.01$ | $0.00 \pm 0.00$ |
| Example2.E2 | $0.41 \pm 0.01$ | $0.38 \pm 0.02$ | $\mathbf{0.00 \pm 0.00}$ | $0.41 \pm 0.01$ | $0.41 \pm 0.01$ | $0.00 \pm 0.00$ |
| **Average** | $0.44 \pm 0.01$ | $0.41 \pm 0.02$ | $\mathbf{0.00 \pm 0.00}$ | $0.45 \pm 0.01$ | $0.45 \pm 0.01$ | $0.00 \pm 0.00$ |
| Example2s.E0 | $0.43 \pm 0.01$ | $0.43 \pm 0.01$ | $\mathbf{0.00 \pm 0.00}$ | $0.43 \pm 0.01$ | $0.43 \pm 0.01$ | $0.00 \pm 0.00$ |
| Example2s.E1 | $0.49 \pm 0.02$ | $0.49 \pm 0.02$ | $\mathbf{0.00 \pm 0.00}$ | $0.49 \pm 0.02$ | $0.49 \pm 0.02$ | $0.00 \pm 0.00$ |
| Example2s.E2 | $0.43 \pm 0.01$ | $0.43 \pm 0.01$ | $\mathbf{0.00 \pm 0.00}$ | $0.43 \pm 0.01$ | $0.43 \pm 0.01$ | $0.00 \pm 0.00$ |
| **Average** | $0.45 \pm 0.01$ | $0.45 \pm 0.01$ | $\mathbf{0.00 \pm 0.00}$ | $0.45 \pm 0.01$ | $0.45 \pm 0.01$ | $0.00 \pm 0.00$ |

## 6.2 IMAGE CLASSIFICATION: COLORED MNIST

For each digit in the MNIST dataset, define $Y = 1$ if the digit is between 0-4 and $Y = 0$ if it is between 5-9. The label of each image is flipped with a probability 0.25 to create the final label. There are three environments in the dataset, i.e., (0.1, 0.2, 0.9), where the number indicates the probability of a digit with label 1 being red. We conducted this experiment using DOMAINBED (Gulrajani & Lopez-Paz, 2020), which provides a standardized and fair testbed for domain generalization. As highlighted in Gulrajani & Lopez-Paz (2020), it is essential for an algorithm to specify the model selection method. In Table 2, we present the comparative results on the Colored MNIST dataset using the leave-one-domain-out cross-validation model selection method. Each column in the table (0.1, 0.2, 0.9) represents a testing environment (other environments were used as training environments). The workflow of FMI is shown in Figure 4

**Conclusion** From Table 2, we can clearly see that FMI surpasses all other methods by a significant margin. Notably, FMI increases the accuracy by more than 20% when the testing environment is 0.9. This is precisely the scenario where the training environments (0.1 and 0.2) contain a strong signal of the spurious feature (color) that is different from testing environment. By matching and emulating perfect intervention on this spurious feature, FMI successfully learns the true feature (digit shape). However, when the training data lacks a strong signal for the spurious feature, FMI shows no advantage over other methods. In such cases, our assumptions are violated. Consequently, with the same number of iterations, FMI might lose some samples in the subsampling process, potentially leading to underperformance. We believe this issue can be resolved with more iterations.

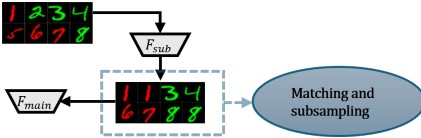

Figure 4: The illustration of FMI workflow with Colored MNIST. Before matching, most digits are green between 0-4 and red between 5-9. After matching, the correlation between color (spurious feature) and digit class (target) disappears.

Table 2: Experimental results on the Colored MNIST dataset in terms of test accuracy for all algorithms. The algorithm that achieves the highest average accuracy and the highest accuracy in environment 0.9 is boldfaced, while the second highest is underlined. The model selection method used is leave-one-domain-out.

| Algorithm | 0.1 | 0.2 | 0.9 | Avg |
|---|---|---|---|---|
| ERM (Vapnik, 1991) | $49.9 \pm 6.1$ | $53.3 \pm 2.2$ | $10.0 \pm 0.0$ | $37.7 \pm 2.3$ |
| IRM (Arjovsky et al., 2019) | $47.0 \pm 3.8$ | $53.0 \pm 2.8$ | $10.0 \pm 0.1$ | $36.7 \pm 1.6$ |
| IB-IRM (Ahuja et al., 2021) | $49.9 \pm 0.2$ | $51.4 \pm 1.1$ | $10.0 \pm 0.1$ | $37.1 \pm 0.4$ |
| IGA (Koyama & Yamaguchi, 2020) | $45.2 \pm 4.5$ | $50.0 \pm 0.6$ | $\underline{31.6 \pm 5.6}$ | $\underline{42.3 \pm 2.4}$ |
| ANDMask (Parascandolo et al., 2020) | $53.7 \pm 2.2$ | $57.0 \pm 3.2$ | $10.1 \pm 0.1$ | $40.3 \pm 1.1$ |
| CORAL (Sun & Saenko, 2016) | $53.9 \pm 4.5$ | $49.6 \pm 0.1$ | $10.0 \pm 0.0$ | $37.8 \pm 1.5$ |
| DANN (Ganin et al., 2016) | $56.1 \pm 4.0$ | $51.9 \pm 2.0$ | $10.1 \pm 0.1$ | $39.4 \pm 1.9$ |
| CDANN (Li et al., 2018b) | $46.5 \pm 6.3$ | $49.3 \pm 0.7$ | $10.2 \pm 0.1$ | $35.4 \pm 2.0$ |
| GroupDRO (Sagawa et al., 2019) | $45.5 \pm 6.0$ | $51.8 \pm 1.5$ | $9.9 \pm 0.1$ | $35.7 \pm 2.1$ |
| MMD (Li et al., 2018a) | $50.1 \pm 0.2$ | $49.9 \pm 0.2$ | $9.9 \pm 0.1$ | $36.6 \pm 0.1$ |
| VREx (Krueger et al., 2021) | $56.8 \pm 3.3$ | $51.9 \pm 2.1$ | $9.9 \pm 0.1$ | $39.6 \pm 1.2$ |
| CausIRL (MMD) (Chevalley et al., 2022) | $47.1 \pm 3.0$ | $53.2 \pm 2.8$ | $10.1 \pm 0.1$ | $36.8 \pm 1.4$ |
| **FMI (OURS)** | $27.0 \pm 5.7$ | $47.5 \pm 2.4$ | $\mathbf{57.9 \pm 4.7}$ | $\mathbf{44.1 \pm 2.2}$ |

**Test the feature learned in training environment** We apply the method in Section 5 on Colored MNIST dataset to test if the feature learned in the training environment is spurious. More specifically, we sampled $n = 200$ images from environment $e = 0.9$, while the training environment is given by $e_0 = 0.1$. The p-values for testing $Y^e | \hat{f}^e = 0$ in the training process are shown in Figure 5. The red dashed line represents significance level $0.05$. As we can see from the figure, in both environments, the feature extracted by FMI passes this goodness-of-fit test with p-value above $0.05$. The feature learned in the training environment, although performs well in the training environment, has extremely small p-value (close to $0$) in the new environment (Figure 5(b)). We conclude that in this example, the feature learned in the training environment is spurious, while FMI extracts the true feature.

### 6.3 IMAGE CLASSIFICATION: WATERBIRDS

We conducted another experiment on a more complicated image dataset – *WaterBirds* (Sagawa et al., 2019), which contains images of birds cut and pasted on different backgrounds. The target of the dataset is to predict whether the bird in the image is water bird or land bird. In this experiment,

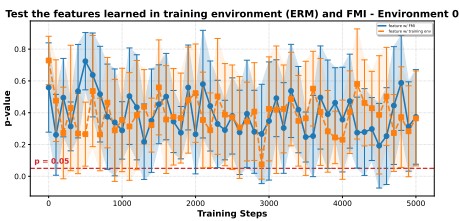 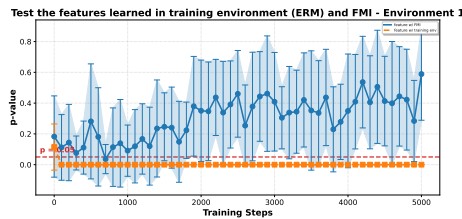

(a) Plot of p-values in the training environment   (b) Plot of p-values in the validation environment

Figure 5: Plots of p-values for testing $Y^e | \hat{f}^e = 0$ in the training environment ($e_0 = 0.1$) and validation environment ($e = 0.9$) of Colored MNIST given different features. In each plot, the feature learned in the training environment is colored orange and the feature learned by FMI is colored blue. The y-axis in each plot represents the p-value of the goodness-of-fit test. The dashed red lines represent significance level $0.05$. The error bars are obtained by repeating the experiment ten times.

we created two environments based on the background of the images and we used those with water background as training environment and test the accuracy on images with land background.

The results, averaged over five runs, are presented in Table 3. Although ERM, trained on the training environment, may capture some of the true features, FMI consistently outperforms other methods, making it a compelling option in practice.

Table 3: Test accuracy of various algorithms on *WaterBirds*. The testing environment consists of images with land backgrounds. The model selection method is training domain validation set. The error bars are calculated by repeating the experiment 5 times.

| Algorithm | Test Accuracy (%) |
|---|---|
| ERM | $77.3 \pm 2.3$ |
| IRM | $73.5 \pm 7.3$ |
| IB-IRM | $73.6 \pm 7.4$ |
| IGA | $72.2 \pm 1.4$ |
| ANDMask | $72.9 \pm 3.4$ |
| CORAL | $77.3 \pm 2.3$ |
| DANN | $76.1 \pm 1.7$ |
| CDANN | $76.1 \pm 1.7$ |
| GroupDRO | $77.3 \pm 2.3$ |
| MMD | $77.3 \pm 2.3$ |
| VREx | $76.8 \pm 2.9$ |
| CausIRL (CORAL) | $75.4 \pm 1.0$ |
| **FMI (OURS)** | $\mathbf{79.3 \pm 1.9}$ |

## 7 Discussion and Future Work

The success of FMI leads us to contemplate the root cause of poor generalizability in domain generalization. From our observations, the generalizability issue primarily arises from the bias of the training data itself. While modern AI models can easily fit any complex functional relationship, they can be 'misled' by the training data. Whenever there is a strong spurious signal in the training data, the model tends to rely on it. However, the intervention suggested by FMI can help the model eliminate the influence of such spurious features. Under these circumstances, FMI manifests superior performance and there may be no need to collect data from multiple domains. It is worth noting that the generalizability issues induced by covariate shift are not due to spurious features. Therefore, it is doubtful whether FMI can be applied to cases where covariate shift is present. As future work, we will explore scenarios where there are multiple spurious features, which could be challenging since emulating perfect interventions on multiple spurious features is not straightforward.

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

APPENDIX

# A   EXPERIMENTS DETAILS

In this work, we mainly relied on two packages – DOMAINBED (Gulrajani & Lopez-Paz, 2020) and linear unit tests (Aubin et al., 2021).

## A.1   DATASETS

We first describe the datasets (Example 2/2S) introduced in Aubin et al. (2021). This example is motivated by Beery et al. (2018) and Arjovsky et al. (2019).

**Example 2/2S.**   Let

$$\mu_{\text{cow}} \sim 1_{d_{\text{inv}}}, \quad \mu_{\text{camel}} = -\mu_{\text{cow}}, \quad \nu_{\text{animal}} = 10^{-2},$$

$$\mu_{\text{grass}} \sim 1_{d_{\text{spu}}}, \quad \mu_{\text{sand}} = -\mu_{\text{grass}}, \quad \nu_{\text{background}} = 1.$$

To construct the datasets $D_e$ for every $e \in \mathcal{E}$ and $i = 1, \ldots, n_e$, sample:

$$j_i^e \sim \text{Categorical}(p^e s^e, (1 - p^e)s^e, p^e(1 - s^e), (1 - p^e)(1 - s^e));$$

$$z_{\text{inv},i}^e \sim \begin{cases} (\mathcal{N}_{d_{\text{inv}}}(0, 10^{-1}) + \mu_{\text{cow}}) \cdot \nu_{\text{animal}} & \text{if} \quad j_i^e \in \{1, 2\}, \\ (\mathcal{N}_{d_{\text{inv}}}(0, 10^{-1}) + \mu_{\text{camel}}) \cdot \nu_{\text{animal}} & \text{if} \quad j_i^e \in \{3, 4\}, \end{cases} \quad z_i^e \leftarrow (z_{\text{inv},i}^e, z_{\text{spu},i}^e);$$

$$z_{\text{spu},i}^e \sim \begin{cases} (\mathcal{N}_{d_{\text{spu}}}(0, 10^{-1}) + \mu_{\text{grass}}) \cdot \nu_{\text{background}} & \text{if} \quad j_i^e \in \{1, 4\}, \\ (\mathcal{N}_{d_{\text{spu}}}(0, 10^{-1}) + \mu_{\text{sand}}) \cdot \nu_{\text{background}} & \text{if} \quad j_i^e \in \{2, 3\}, \end{cases} \quad y_i^e \leftarrow \begin{cases} 1 & \text{if} \quad 1_{d_{\text{inv}}}^\top z_{i,\text{inv}}^e > 0, \\ 0 & \text{else}; \end{cases}$$

$$x_i^e \leftarrow S(z_i^e),$$

(6)

where the environment foreground/background probabilities are $p^{e=E_0} = 0.95$, $p^{e=E_1} = 0.97$, $p^{e=E_2} = 0.99$ and the cow/camel probabilities are $s^{e=E_0} = 0.3$, $s^{e=E_1} = 0.5$, $s^{e=E_2} = 0.7$. For $n_{env} > 3$ and $j \in [3 : n_{\text{env}} - 1]$, the extra environment variables are respectively drawn according to $p^{e=E_j} \sim \text{Unif}(0.9, 1)$ and $s^{e=E_j} \sim \text{Unif}(0.3, 0.7)$. The scrabling matrix $S$ is set to identity in Example 2 and a random unitary matrix is selected to rotate the latents in Example 2S.

This Example corresponds to Figure 1(a).

Next, we introduce the Colored MNIST dataset we used.

**Colored MNIST**   We follow the construction in DOMAINBED (Gulrajani & Lopez-Paz, 2020), where the task is binary classification – identify whether the digit is less than 5 (not including 5) or more than 5. There are three environments: two training environments containing 25,000 images each and one test environment containing 10,000 images. Define $Y = 1$ if the digit is between 0-4 and $Y = 0$ if it is between 5-9. The label of each image is flipped with probability 0.25 as final label. The spurious feature $Z_{spu}^e$ in each environment is obtained by flipping the final label with certain probability corresponding to each environment. For the three environments, we index them by $e = 0.1, 0.2, 0.9$, each representing the probability of flipping final label to obtain the spurious feature. Finally, if $Z_{spu}^e = 1$, we color the digit green, otherwise, we color it red. For this dataset, spurious feature is color and the true feature is the shape of the digit. To visualize this dataset, we transform it to shape $(3, 28, 28)$.

**WaterBirds**   We downloaded *WaterBirds* dataset following the instructions given by Sagawa et al. (2019) and then divide the dataset set into two environments according to the background type. After importing the dataset, we transform all the images to shape $(3, 224, 224)$. See Figure 6 for an example of images from the dataset.

## A.2   TRAINING PROCEDURE

**Example 2/2S**   We followed the same setting as in Aubin et al. (2021). We use random hyperparameter search and use 2 hyperparameter queries and average over 10 data seeds. For Example 2/2S, we generated 1000 samples each time and run each algorithm for 10000 iterations (each iteration use the full data and the two networks of FMI are trained together). The evaluation of the performance on Example 2/2S are reported using the classification errors and standard deviations.

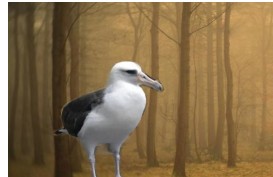

     (a) Background: water           (b) Background: land

Figure 6: Example images from *WaterBirds*. The environments in this dataset are water and land backgrounds. The labels are waterbird and landbird. We use images with water backgrounds as the training environment.

**Colored MNIST** As in DOMAINBED (Gulrajani & Lopez-Paz, 2020), the network is separated into featurizer and classifier. For the featurizer, we used the default CNN architecture from DO-MAINBED. There are four convolutional layers with feature map dimensions 64, 128, 128, 128. Each convolutional layer is followed by a ReLU activation and group normalization layer. The final output layer of the CNN is an average pooling layer with output size 128. For the classifier, we used an MLP architecture with three fully connected layers, with output sizes 64, 32, 2. The prediction of the neural network were based on the last layer of the classifier. The hyperparameter search is in accordance with DOMAINBED. In our experiment, we ran each algorithm 10 times with default hyperparameter. For FMI, we chose batch size to be 64, and conduct subsampling each time we collect at least 32 inputs in each predicted group. For the evaluation, we reported accuracy and standard deviations (averaged over ten trials except IGA, which is averaged over eight trials). We tried all model selection methods given in DOMAINBED. More experimental results can be found in Section A.3.

**WaterBirds** As in DOMAINBED (Gulrajani & Lopez-Paz, 2020), the network is separated into featurizer and classifier. For the featurizer, we used the default ResNet18 architecture from DO-MAINBED. For the classifier, we used an MLP architecture with three fully connected layers, with output sizes 64, 32, 2. The prediction of the neural network were based on the last layer of the classifier. The hyperparameter search is in accordance with DOMAINBED. In our experiment, we ran each algorithm 5 times with default hyperparameter. For FMI, we chose batch size to be 64, and conduct subsampling each time we collect at least 32 inputs in each predicted group. For the evaluation, we reported accuracy and standard deviations. The model selection method is training domain validation set.

## A.3 SUPPLEMENTARY EXPERIMENTS

**Colored MNIST** In Table 4 and Table 5, we provide the supplementary experiments for Colored MNIST with a different model selection methods , i.e., training-domain validation set and test-domain validation set (oracle), which are specified in Gulrajani & Lopez-Paz (2020).

**Learn two features together v.s. Learn spurious feature first.** We provide the supplementary experiments to study the difference of training strategies. Previously in all our experiments, we train two neural networks together for 5000 steps. One of them is used to learn the spurious feature, which is called the subnetwork. Then, we subsample from the training data based on the subnetwork and (5) and then use this subsample to train the other network (the main network). However, we can also train the subnetwork for enough steps and then train the other network while fixing the subnetwork. We tried three strategies in this experiment:

1. Train subnetwork and the main network together for 5,000 steps. In each step, we update both subnetwork and main network and use the classification result of the subnetwork to conduct subsampling;
2. Train subnetwork for 4,000 steps to warm up. Then we use the classification result of the subnetwork to conduct subsampling and train the main network for 4,000 steps;
3. Train subnetwork for 5,000 steps to warm up. Then we use the classification result of the subnetwork to conduct subsampling and train the main network for 5,000 steps;

Table 4: Experimental results on Colored MNIST in terms of test accuracy for all algorithms. The algorithm that achieves the highest average accuracy and the highest accuracy in environment 0.9 is boldfaced, while the second highest is underlined. The model selection method used is training-domain validation set.

| Algorithm | 0.1 | 0.2 | 0.9 | Avg |
|---|---|---|---|---|
| ERM | $71.6 \pm 0.2$ | $72.9 \pm 0.1$ | $10.3 \pm 0.1$ | $51.6 \pm 0.1$ |
| IRM | $65.2 \pm 1.2$ | $63.7 \pm 0.8$ | $10.0 \pm 0.1$ | $46.3 \pm 0.5$ |
| IB-IRM | $65.0 \pm 1.4$ | $68.7 \pm 0.8$ | $10.0 \pm 0.1$ | $47.9 \pm 0.5$ |
| IGA | $45.1 \pm 4.5$ | $50.3 \pm 0.6$ | $\underline{22.4 \pm 5.6}$ | $39.2 \pm 2.1$ |
| ANDMask | $71.3 \pm 0.2$ | $73.2 \pm 0.1$ | $10.2 \pm 0.1$ | $51.6 \pm 0.1$ |
| CORAL | $71.6 \pm 0.1$ | $73.0 \pm 0.1$ | $10.1 \pm 0.1$ | $51.6 \pm 0.1$ |
| DANN | $71.9 \pm 0.1$ | $73.0 \pm 0.1$ | $10.2 \pm 0.1$ | $51.7 \pm 0.1$ |
| CDANN | $72.7 \pm 0.2$ | $73.2 \pm 0.2$ | $10.2 \pm 0.1$ | $\mathbf{52.1 \pm 0.1}$ |
| GroupDRO | $72.9 \pm 0.1$ | $73.0 \pm 0.2$ | $10.1 \pm 0.1$ | $52.0 \pm 0.1$ |
| MMD | $51.2 \pm 0.4$ | $52.4 \pm 1.0$ | $10.0 \pm 0.1$ | $37.9 \pm 0.4$ |
| VREx | $72.8 \pm 0.2$ | $73.3 \pm 0.1$ | $10.0 \pm 0.1$ | $\underline{52.0 \pm 0.0}$ |
| CausIRL (MMD) | $48.2 \pm 3.1$ | $52.6 \pm 0.9$ | $10.0 \pm 0.1$ | $37.0 \pm 1.2$ |
| **FMI (OURS)** | $22.3 \pm 2.9$ | $51.3 \pm 1.6$ | $\mathbf{28.0 \pm 6.0}$ | $33.8 \pm 2.3$ |

Table 5: Experimental results on Colored MNIST in terms of test accuracy for all algorithms. The algorithm that achieves the highest average accuracy and the highest accuracy in environment 0.9 is boldfaced, while the second highest is underlined. The model selection method used is test-domain validation set.

| Algorithm | 0.1 | 0.2 | 0.9 | Avg |
|---|---|---|---|---|
| ERM | $63.4 \pm 0.3$ | $67.9 \pm 0.2$ | $24.1 \pm 0.8$ | $51.8 \pm 0.3$ |
| IRM | $58.4 \pm 1.8$ | $57.6 \pm 1.2$ | $49.8 \pm 0.2$ | $\underline{55.3 \pm 0.9}$ |
| IB-IRM | $52.9 \pm 2.0$ | $56.2 \pm 2.5$ | $33.1 \pm 5.6$ | $47.4 \pm 2.3$ |
| IGA | $50.1 \pm 0.1$ | $50.3 \pm 0.2$ | $\underline{50.4 \pm 0.1}$ | $50.2 \pm 0.1$ |
| ANDMask | $69.0 \pm 0.5$ | $72.6 \pm 0.2$ | $18.4 \pm 1.0$ | $53.3 \pm 0.4$ |
| CORAL | $63.6 \pm 0.8$ | $67.5 \pm 0.5$ | $25.3 \pm 1.0$ | $52.1 \pm 0.3$ |
| DANN | $70.3 \pm 0.4$ | $71.9 \pm 0.3$ | $18.0 \pm 1.5$ | $53.4 \pm 0.5$ |
| CDANN | $72.3 \pm 0.4$ | $72.7 \pm 0.2$ | $16.0 \pm 0.9$ | $53.7 \pm 0.3$ |
| GroupDRO | $65.7 \pm 0.5$ | $67.4 \pm 0.5$ | $31.9 \pm 1.5$ | $55.0 \pm 0.4$ |
| MMD | $50.3 \pm 0.2$ | $51.2 \pm 0.5$ | $10.6 \pm 0.5$ | $37.4 \pm 0.3$ |
| VREx | $69.8 \pm 0.5$ | $72.2 \pm 0.1$ | $24.9 \pm 1.3$ | $\mathbf{55.7 \pm 0.4}$ |
| CausIRL (MMD) | $50.3 \pm 0.2$ | $50.4 \pm 0.1$ | $10.3 \pm 0.2$ | $37.0 \pm 0.1$ |
| **FMI (OURS)** | $22.4 \pm 5.1$ | $50.8 \pm 4.3$ | $\mathbf{60.9 \pm 2.6}$ | $44.7 \pm 2.4$ |

Below in Table 6, we show the comparison of different training strategies. In general, strategy 1 gives better results.

Table 6: Performance of FMI on Colored MNIST with different training strategy

| Strategy | Model Selection Method | 0.1 | 0.2 | 0.9 | Avg |
|---|---|---|---|---|---|
| Strategy 1 | Training-domain validation set | $22.3 \pm 2.9$ | $51.3 \pm 1.6$ | $28.0 \pm 6.0$ | $33.8 \pm 2.3$ |
| | Leave-one-domain-out cross-validation | $27.0 \pm 5.7$ | $47.5 \pm 2.4$ | $57.9 \pm 4.7$ | $44.1 \pm 2.2$ |
| | Test-domain validation set (oracle) | $22.4 \pm 5.1$ | $50.8 \pm 4.3$ | $60.9 \pm 2.6$ | $44.7 \pm 2.4$ |
| Strategy 2 | Training-domain validation set | $26.2 \pm 5.3$ | $53.7 \pm 0.9$ | $10.4 \pm 0.4$ | $30.1 \pm 1.8$ |
| | Leave-one-domain-out cross-validation | $29.7 \pm 5.4$ | $53.8 \pm 2.7$ | $17.1 \pm 1.9$ | $33.5 \pm 2.3$ |
| | Test-domain validation set (oracle) | $43.1 \pm 2.2$ | $51.7 \pm 1.1$ | $15.6 \pm 0.8$ | $36.8 \pm 0.9$ |
| Strategy 3 | Training-domain validation set | $58.6 \pm 1.3$ | $69.3 \pm 0.6$ | $10.1 \pm 0.1$ | $46.0 \pm 0.5$ |
| | Leave-one-domain-out cross-validation | $45.0 \pm 2.4$ | $47.0 \pm 4.5$ | $22.5 \pm 4.6$ | $38.2 \pm 2.7$ |
| | Test-domain validation set (oracle) | $57.4 \pm 1.5$ | $68.6 \pm 0.6$ | $11.6 \pm 0.6$ | $45.8 \pm 0.7$ |

**FMI when there is single training environment** In the following tables, we show the testing accuracy of FMI when there is only one training environment in ColoredMNIST. In Table 7, The training environment we use is $e = 0.1$ and the testing environment is $e = 0.9$. In Table 8, the training environment we use is $e = 0.05$ and the testing environment is $e = 0.95$, which is more imbalanced compared to previous setting. Notice that the difference of testing accuracy across different model selection methods is huge. In fact, the model selection method is crucial in the experiment. Nevertheless, under any model selection method, FMI surpasses others by a large margin.

Table 7: Experimental results on the Colored MNIST dataset in terms of test accuracy for all algorithms (two environments). Each column represents a different model selection method. The training environment here is 0.1 and the testing environment is 0.9

| Algorithm | Training-domain validation set | Test-domain validation set |
|---|---|---|
| ERM | $10.1 \pm 0.1$ | $12.1 \pm 0.8$ |
| IRM | $10.0 \pm 0.0$ | $10.0 \pm 0.0$ |
| IB-IRM | $10.0 \pm 0.0$ | $42.0 \pm 4.2$ |
| IGA | $10.0 \pm 0.1$ | $11.3 \pm 0.5$ |
| ANDMask | $10.0 \pm 0.0$ | $11.0 \pm 0.2$ |
| CORAL | $10.0 \pm 0.1$ | $11.2 \pm 0.5$ |
| DANN | $9.9 \pm 0.1$ | $10.0 \pm 0.1$ |
| CDANN | $9.9 \pm 0.1$ | $10.0 \pm 0.1$ |
| GroupDRO | $10.0 \pm 0.0$ | $11.3 \pm 0.4$ |
| MMD | $10.1 \pm 0.1$ | $12.1 \pm 0.8$ |
| VREx | $10.0 \pm 0.0$ | $12.3 \pm 0.5$ |
| CausIRL (MMD) | $10.0 \pm 0.0$ | $10.0 \pm 0.0$ |
| **FMI (OURS)** | $\mathbf{30.7 \pm 5.9}$ | $\mathbf{71.1 \pm 0.3}$ |

Table 8: Experimental results on the Colored MNIST dataset in terms of test accuracy for all algorithms (two environments). Each column represents a different model selection method. The training environment here is 0.05 and the testing environment is 0.95

| Algorithm | Training-domain validation set | Test-domain validation set |
|---|---|---|
| ERM | $5.0 \pm 0.0$ | $5.1 \pm 0.1$ |
| IRM | $5.0 \pm 0.0$ | $5.0 \pm 0.0$ |
| IB-IRM | $5.0 \pm 0.0$ | $45.6 \pm 4.3$ |
| IGA | $5.0 \pm 0.0$ | $5.0 \pm 0.0$ |
| ANDMask | $5.0 \pm 0.0$ | $5.1 \pm 0.0$ |
| CORAL | $5.0 \pm 0.0$ | $5.1 \pm 0.1$ |
| DANN | $4.9 \pm 0.0$ | $4.9 \pm 0.0$ |
| CDANN | $4.9 \pm 0.0$ | $4.9 \pm 0.0$ |
| GroupDRO | $5.0 \pm 0.0$ | $5.1 \pm 0.1$ |
| MMD | $5.0 \pm 0.0$ | $5.0 \pm 0.0$ |
| VREx | $5.0 \pm 0.0$ | $5.1 \pm 0.1$ |
| CausIRL (MMD) | $5.0 \pm 0.0$ | $5.0 \pm 0.0$ |
| **FMI (OURS)** | $\mathbf{21.4 \pm 7.3}$ | $\mathbf{69.1 \pm 0.5}$ |

**Does FMI extract the true feature?** Although FMI demonstrates superior performance, it remains a question whether the feature extracted by FMI is the true feature, i.e., the shape of the digit. To address this, we applied Grad-CAM (Selvaraju et al., 2017) to visualize the features of the CNN used in this experiment. Figure 7 shows a comparison of features extracted by FMI and ERM. The models producing the figure are FMI and ERM, trained in environments (0.1, 0.2), and the images

are sampled from environment 0.9. ERM, with low accuracy, focuses on areas irrelevant to the shape of the digit, whereas FMI concentrates on distinctive parts of the digits (e.g., the ○ parts in 6 and 8).



(a) Attention map of FMI                    (b) Attention map of ERM

Figure 7: Attention maps of FMI and ERM obtained by Grad-CAM. The highlighted areas are what our model used to predict the class of the image.

## A.4 COMPUTE DESCRIPTION

Our computing resource is one Tesla V100-SXM2-16GM with 16 CPU cores.

## B   PROOF OF THEOREM 1

We restate Theorem 1 for convenience.

**Theorem 2.** *Under Assumptions 1-4, any solution to (FMI) achieves the minimax risk as in Formula (2). Therefore, FMI offers OOD generalization.*

Before we prove Theorem 1, we need the following lemma:

**Lemma 1.** *With $\ell(\cdot)$ being zero-one loss, the lowest error of any linear classifier that is achievable in any environment is $q$.*

**Proof of Lemma 1.**   This proof is similar to the proof of Ahuja et al. (2021)[Theorem 4].

Consider any environment $e$, for any $\Phi \in \mathbb{R}^{(m+o)}$, we have the following decomposition

$$\Phi \cdot X = \Phi \cdot S(Z_{true}, Z_{spu}) = \Phi_{true} \cdot Z_{true} + \Phi_{spu} \cdot Z_{spu}. \tag{7}$$

Let

$$\mathcal{Z}_+^{(e)} = \{(z_{true}^{(e)}, z_{spu}^{(e)}) : \mathbb{I}(\Phi_{true} \cdot z_{true}^{(e)} + \Phi_{spu} \cdot z_{spu}^{(e)}) = \mathbb{I}(w_{true} \cdot z_{true}^{(e)})\} \tag{8}$$

$$\mathcal{Z}_-^{(e)} = \{(z_{true}^{(e)}, z_{spu}^{(e)}) : \mathbb{I}(\Phi_{true} \cdot z_{true}^{(e)} + \Phi_{spu} \cdot z_{spu}^{(e)}) \neq \mathbb{I}(w_{true} \cdot z_{true}^{(e)})\} \tag{9}$$

and assume $P((Z_{true}^{(e)}, Z_{spu}^{(e)}) \in \mathcal{Z}_+^{(e)}) = p$. By definition of the risk, we have

$$\mathcal{R}^e(\Phi) = \mathbb{E}\left[\mathbb{I}(w_{true} \cdot Z_{true}^e) \oplus N^e \oplus \mathbb{I}(\Phi_{true} \cdot Z_{true}^e + \Phi_{spu} \cdot Z_{spu}^e)\right] \tag{10}$$

$$= p\mathbb{E}(1 \oplus N^e) + (1-p)\mathbb{E}(N^e) > q. \tag{11}$$

$\square$

The following Corollary gives the structure of the optimal classifier:

**Corollary 1.** *In any environment $e \in \mathcal{E}$, the optimal predictor $\mathbb{I}(\Phi^e)$ should agree with $\mathbb{I}(w_{true} \cdot z_{true}^e)$ everywhere in the support.*

**Proof of Corollary 2.**   Check (13).                                                          $\square$

Next, we prove the following lemma:

**Lemma 2.** *Suppose $x, \alpha \in \mathbb{R}^m, y, \beta, \gamma \in \mathbb{R}^o$ and $\|\beta\| < 1$, then*

$$f(x, y) = x^T \alpha \gamma^T y + y^T \beta \gamma^T y \geq 0 \tag{12}$$

*within some bounded ball centered at zero if and only if $\alpha = 0$ and $\beta \gamma^T$ is positive semi-definite.*

**Proof of Lemma 2.**   Without loss of generality, assume the radius of the bounded ball is 1.

If $\alpha = 0$ and $\beta \gamma^T$ is positive semi-definite, then $f(x, y) = y^T \beta \gamma^T y \geq 0$.

Now, assume $f(x, y) \geq 0$. If $\alpha \neq 0$, assume $\gamma^T \beta \geq 0$, then take $x = c\alpha/\|\alpha\|^2$ and $y = -\gamma/\|\gamma\|^2$ with $\gamma^T \beta/\|\gamma\|^2 < c < 1$. We have

$$f(x, y) = \frac{c}{\|\alpha\|^2} \alpha^T \alpha \gamma^T \frac{-\gamma}{\|\gamma\|^2} + \frac{-\gamma^T}{\|\gamma\|^2} \beta \gamma^T \frac{-\gamma}{\|\gamma\|^2} \tag{13}$$

$$= -c + \frac{\gamma^T \beta}{\|\gamma\|^2} < 0 \tag{14}$$

We can similarly prove the case when $\gamma^T \beta < 0$. Therefore, we know $\alpha = 0$ and therefore $\beta \gamma^T$ must be positive semi-definite.                                                          $\square$

With these in mind, we can prove Theorem 1.

**Proof of Theorem 1.** By Corollary 1, we know the solution $\Phi^{e_m}$ to (FMI) must agree with $\mathbb{I}(w_{true} \cdot z_{true}^{e_m})$. Suppose

$$\Phi^{e_m} \cdot X = \Phi^{e_m} \cdot S(Z_{true}, Z_{spu}) = \Phi_{true}^{e_m} \cdot Z_{true} + \Phi_{spu}^{e_m} \cdot Z_{spu}, \tag{15}$$

it holds that

$$(\Phi_{true}^{e_m} \cdot z_{true} + \Phi_{spu}^{e_m} \cdot z_{spu}) \cdot (w_{true} \cdot z_{true}) \geq 0, \tag{16}$$

for any $z_{true}, z_{spu}$ in the support. In order to use Lemma 2, we normalize $\Phi_{true}^{e_m}$ such that $\|\Phi_{true}^{e_m}\| < 1$, this can be done by transforming $\Phi_{true}^{e_m}$ and $z_{true}$ together.

Now, since $Z_{spu}^{e_m} \perp\!\!\!\perp Y^{e_m}$ by definition and $Y^{e_m} \leftarrow \mathbb{I}(w_{true} \cdot Z_{true}) \oplus \mathcal{N}^e$, we know $Z_{spu}^{e_m} \perp\!\!\!\perp Z_{true}^{e_m}$. Hence, $\mathrm{supp}(z_{true}^{e_m}, z_{spu}^{e_m}) = \mathrm{supp}(z_{true}^{e_m}) \times \mathrm{supp}(z_{spu}^{e_m})$ and by assumption, it contains the unit ball.

Further note that (19) is equivalent to

$$z_{spu}^T \Phi_{spu}^{e_m} w_{true}^T z_{true} + z_{true}^T \Phi_{true}^{e_m} w_{true}^T z_{true} \geq 0. \tag{17}$$

By Lemma 2, we know $\Phi_{spu}^{e_m} = 0$. Thus, the FMI solution uses only the true feature and is mini-max optimal. $\qquad\square$

# C INVARIANCE PRINCIPLE

## C.1 INVARIANCE PRINCIPLE AND IRM

Invariance principle was defined in Arjovsky et al. (2019, Definition 3).

**Definition 3.** *We say that a data representation* $\Phi : \mathcal{X} \to \mathcal{H}$ *elicits an invariant predictor* $w \circ \Phi$ *across environments* $\mathcal{E}$ *if there is a classifier* $w : \mathcal{H} \to \mathcal{Y}$ *simultaneously optimal for all environments, that is,* $w \in \arg\min_{\bar{w}:\mathcal{H}\to\mathcal{Y}} R^e(\bar{w} \circ \Phi)$ *for all* $e \in \mathcal{E}$.

We can see from the definition that whenever there is only one environment in the training dataset, the definition becomes nothing but minimizing the risk in the training data. Therefore, invariance principle makes no effect on the classifier in that case.

Also, IRM requires the environment lie in a linear general position (Arjovsky et al., 2019, Assumption 8), which is formally defined as follows:

**Assumption 5.** *A set of training environments* $\mathcal{E}_{\text{tr}}$ *lie in* linear general position *of degree* $r$ *if* $|\mathcal{E}_{\text{tr}}| > d - r + \frac{d}{r}$ *for some* $r \in \mathbb{N}$, *and for all non-zero* $x \in \mathbb{R}^d$:

$$\dim\left(\text{span}\left(\left\{\mathbb{E}_{X^e}\left[X^e X^{e\top}\right] x - \mathbb{E}_{X^e,\epsilon^e}\left[X^e \epsilon^e\right]\right\}_{e\in\mathcal{E}_{\text{tr}}}\right)\right) > d - r.$$

This assumption limites the exent to which the training environments are co-linear. Under this assumption, the feature learned by IRM can be shown to generalize to all environments.

## C.2 FULLY INFORMATIVE INVARIANT FEATURES AND PARTIALLY INFORMATIVE INVARIANT FEATURES

In Ahuja et al. (2021), the authors categorized invariant features $\Phi^*(\cdot)$ into two types: fully informative invariant features (FIIF) and partially informative invariant features (PIIF).

- FIIF: $\forall e \in \mathcal{E}, Y^e \perp\!\!\!\perp X^e | \Phi^*(X^e)$;
- PIIF: $\exists e \in \mathcal{E}, Y^e \not\perp\!\!\!\perp X^e | \Phi^*(X^e)$.

For different types of features, they gave different theoretical results on whether IRM fail or not.

# D  BACKGROUND ON STRUCTURAL CAUSAL MODELS AND INTERVENTIONS

For completeness, we provide a more detailed background on structural causal models (SCMs) and interventions. This section is borrowed from Peters et al. (2017, Chapter 6).

First, we provide the defintion of SCM and its entailed distribution.

**Definition 4** (Structural causal models). *A structural causal model (SCM) $\mathfrak{C} := (\mathbf{S}, P_{\mathbf{N}})$ consists of a collection $\mathbf{S}$ of d (structural) assignments*

$$X_j := f_j(\mathbf{PA}_j, N_j), \quad j = 1, \ldots, d, \tag{18}$$

*where $\mathbf{PA}_j \subset \{X_1, \ldots, X_d\} \backslash \{X_j\}$ are called parents of $X_j$; and a joint distribution $P_{\mathbf{N}} = P_{N_1, \ldots N_d}$ over the noise variables, which we require to be jointly independent; that is, $P_{\mathbf{N}}$ is a product distribution.*

*The graph $\mathcal{G}$ of an SCM is obtained by creating one vertex for each $X_j$ and drawing directed edges from each parent in $\mathbf{PA}_j$ to $X_j$, that is, from each variable $X_k$ occurring on the right-hand side of equation (21) to $X_j$. We henceforth assume this graph to be acyclic.*

*We sometimes call the elements of $\mathbf{PA}_j$ not only parents but also direct causes of $X_j$, and we call $X_j$ a direct effect of each of its direct causes. SCMs are also called (nonlinear) SEMs.*

**Definition 5** (Entailed distributions). *An SCM $\mathfrak{C}$ defines a unique distribution over the variables $\mathbf{X} = (X_1, \ldots, X_d)$ such that $X_j = f_j(\mathbf{PA}_j, N_j)$, in distribution, for $j = 1, \ldots, d$. We refer to it as the entailed distribution $P_{\mathbf{X}}^{\mathfrak{C}}$ and sometimes write $P_{\mathbf{X}}$.*

Next, we can define intervention on SCM.

**Definition 6** (Intervention distribution). *Consider an SCM $\mathfrak{C} = (\mathbf{S}, P_{\mathbf{N}})$ and its entailed distribution $P_{\mathbf{X}}^{\mathfrak{C}}$. We replace one (or several) of the structural assignments to obtain a new SCM $\tilde{\mathfrak{C}}$. Assume that we replace the assignment for $X_k$ by*

$$X_k := \tilde{f}(\widetilde{\mathbf{PA}}_k, \tilde{N}_k).$$

*We then call the entailed distribution of the new SCM an intervention distribution and say that the variables whose structural assignment we have replaced have been **intervened** on. We denote the new distribution by*

$$P_{\mathbf{X}}^{\tilde{\mathfrak{C}}} := P_{\mathbf{X}}^{\mathfrak{C};do(X_k := \tilde{f}(\widetilde{\mathbf{PA}}_k, \tilde{N}_k))}.$$

*The set of noise variables in $\tilde{\mathfrak{C}}$ now contains both some "new" $\tilde{N}$'s and some "old" $N$'s, all of which are required to be jointly independent.*

*When $\tilde{f}(\widetilde{\mathbf{PA}}_k, \tilde{N}_k)$ puts a point mass on a real value $a$, we simply write $P_{\mathbf{X}}^{\mathfrak{C};do(\tilde{X}_k := a)}$ and call this an **atomic** intervention. When $\tilde{f}(\widetilde{\mathbf{PA}}_k, \tilde{N}_k)$ is a exogenous random variable $\epsilon_k$, we call this a **stochastic** intervention. Atomic intervention, together with stochastic intervention, are called **perfect** intervention. An intervention with $\widetilde{\mathbf{PA}}_k = \mathbf{PA}_k$, that is, where direct causes remain direct causes, is called **imperfect**.*

*We require that the new SCM $\tilde{\mathfrak{C}}$ have an acyclic graph; the set of allowed interventions thus depends on the graph induced by $\tilde{\mathfrak{C}}$.*

# E  GOODNESS-OF-FIT TEST FOR TESTING THE FEATURE LEARNED IN THE TRAINING ENVIRONMENT

In this section, we introduce the specific hypothesis test used in this paper to check whether the feature learned in the training environment is spurious or not. By Proposition 1, we have the following null hypothesis given a environment $e \neq e_0$:

$$H_0 : Y^e | Z_{\text{spu}}^e \stackrel{d}{=} Y^{e_0} | Z_{\text{spu}}^{e_0}.$$

Since the classifier $\hat{f}$ maps different regions of the support of $Z_{\text{spu}}$ into different values, we can approximate $H_0$ by the following hypotheses:

$$H_0^k : Y^e | \hat{f}^e = k \stackrel{d}{=} Y^{e_0} | \hat{f}^{e_0} = k, \quad k = 1, 2, \ldots, K,$$

where $K$ is the number of classes in the problem.

Now, for each $H_0^k$, the distributions we are testing are discrete. Therefore, we can use traditional chi square goodness-of-fit test.

In practice, we can use a sample from $Y^e | \hat{f}^e = k$ and $Y^{e_0} | \hat{f}^{e_0} = k$ to conduct the test in order to make it more efficient. In our experiment, we sampled $n = 200$ images from each distribution and calculated the p-value using chi square distribution. Below in Figure 7, we include the p-values of testing $Y^e | \hat{f}^e = 1$ in our Colored MNIST example. Again, we can clearly see that $e = 0.9$ is a validation environment that rejects the feature learned in the training environment, while the feature learned by FMI remains valid.

We also include the plots for p-values in testing on *WaterBirds* dataset here. See Figure 9 and Figure 10.

Additionally, for the ColoredMNIST experiment, if we use the mixture of $e = 0.2$ and $e = 0.9$ as training environments, we would get a good enough feature based on ERM, as shown in Figure 11, Figure 12 and Figure 13. In this case, there is no need to conduct FMI.

When we have only one training environment and the validation environment is relatively similar to the training environment, as we will show in the following two experiments, the test usually would not reject the feature learned in the training environment through ERM.

In Figure 14 and Figure 15, we demonstrate the test results when $e = 0.6$ is training environment and $e = 0.4$ is testing environment.

In Figure 16 and Figure 17, we demonstrate the test results when $e = 0.8$ is training environment and $e = 0.7$ is testing environment.

As we can see, the feature learned in the training environment through ERM in both experiments cannot be rejected, and our method suggests the feature learned directly through ERM is good enough based on the data at hand.

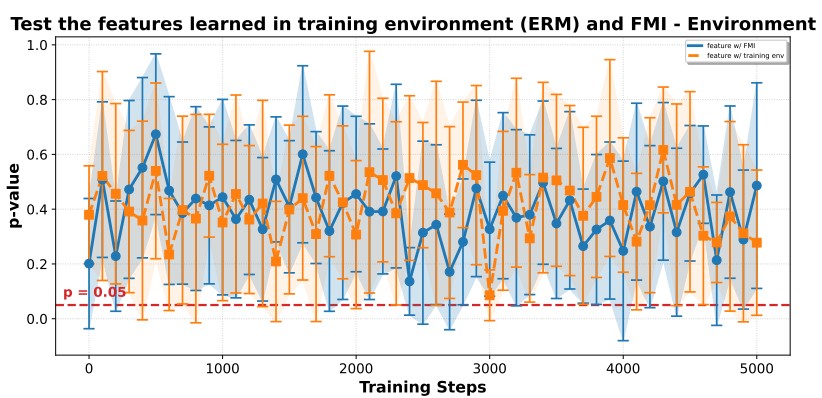

(a) Plot of p-values in the training environment

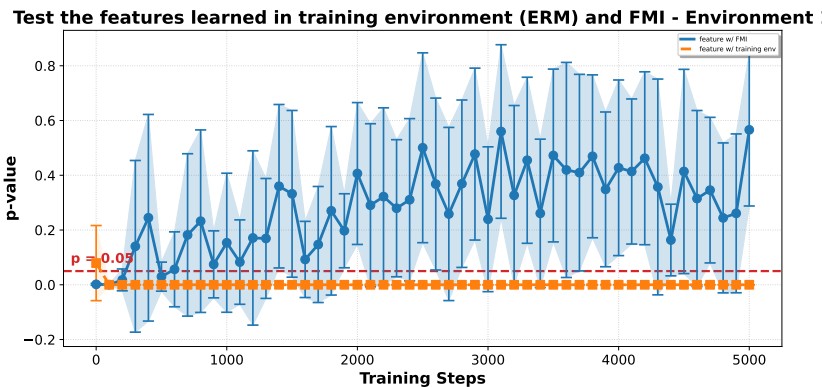

(b) Plot of p-values in the validation environment

Figure 8: Plots of p-values for testing $Y^e | \hat{f}^e = 1$ in the training environment and validation environment of Colored MNIST given different features. In each plot, the feature learned in the training environment is colored orange and the feature learned by FMI is colored blue. The y-axis in each plot represents the p-value of the goodness-of-fit test. The dashed red lines represent significance level $0.05$. The error bars are obtained by repeating the experiment ten times.

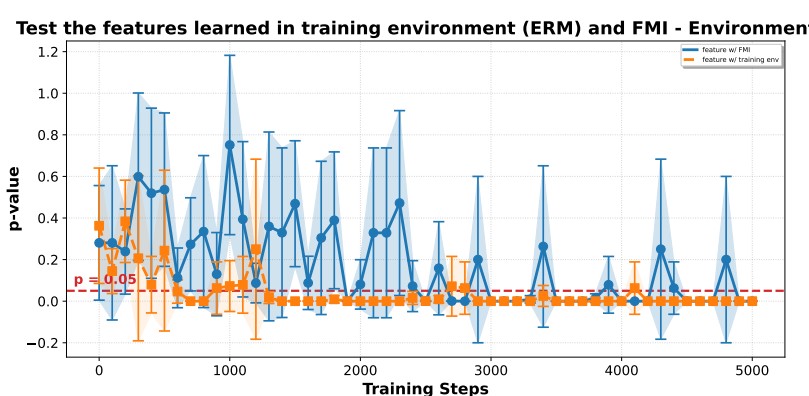

(a) Plot of p-values in the training environment

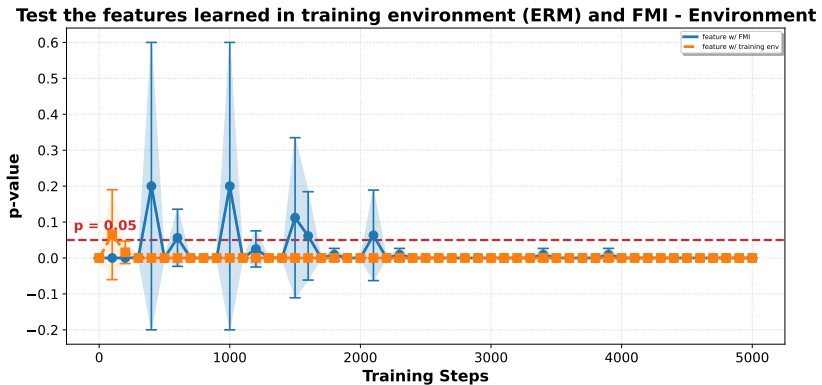

(b) Plot of p-values in the validation environment

Figure 9: Plots of p-values for testing $Y^e | \hat{f}^e = 0$ in the training environment and validation environment of *WaterBirds* given different features. In each plot, the feature learned in the training environment is colored orange and the feature learned by FMI is colored blue. The y-axis in each plot represents the p-value of the goodness-of-fit test. The dashed red lines represent significance level 0.05. The error bars are obtained by repeating the experiment five times.

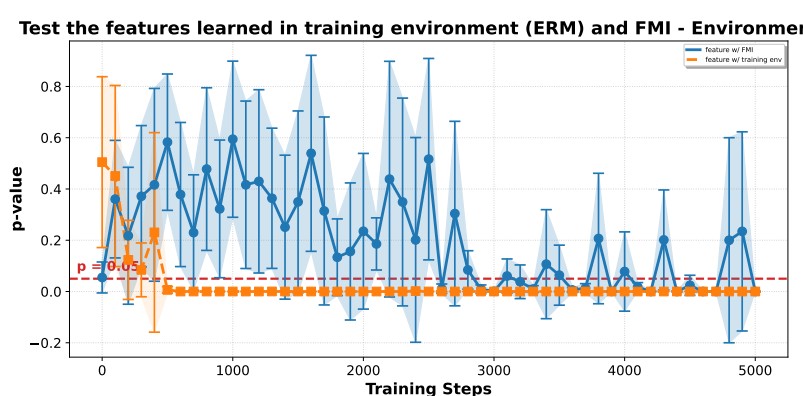

(a) Plot of p-values in the training environment

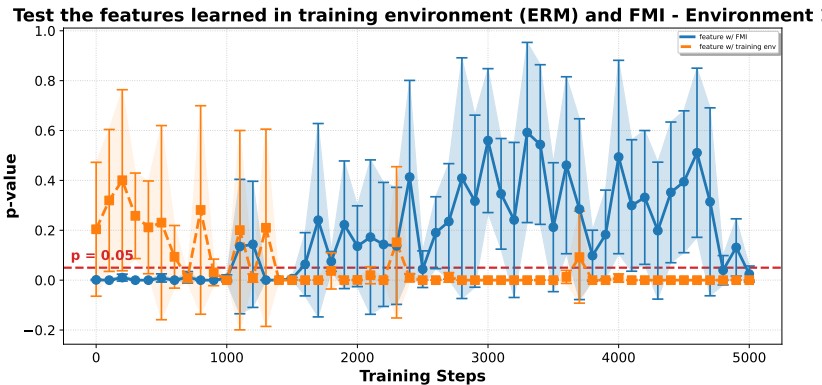

(b) Plot of p-values in the validation environment

Figure 10: Plots of p-values for testing $Y^e | \hat{f}^e = 1$ in the training environment and validation environment of *WaterBirds* given different features. In each plot, the feature learned in the training environment is colored orange and the feature learned by FMI is colored blue. The y-axis in each plot represents the p-value of the goodness-of-fit test. The dashed red lines represent significance level $0.05$. The error bars are obtained by repeating the experiment five times.

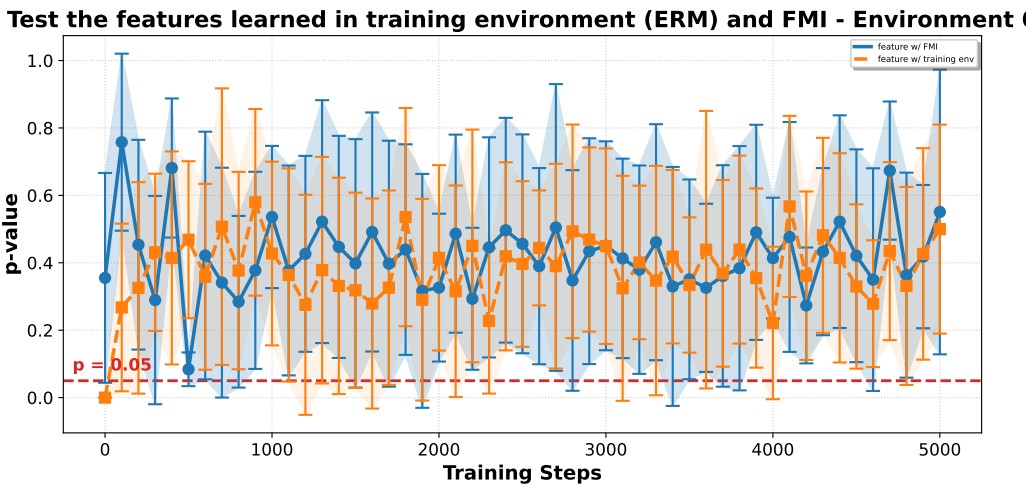

Figure 11: Plots of p-values for testing $Y^e | \hat{f}^e = 1$ in the environment $e = 0.1$ of Colored MNIST given different features. In each plot, the feature learned in the training environment ($e = 0.9$) is colored orange and the feature learned by FMI is colored blue. The y-axis in each plot represents the p-value of the goodness-of-fit test. The dashed red lines represent significance level 0.05. The error bars are obtained by repeating the experiment ten times.

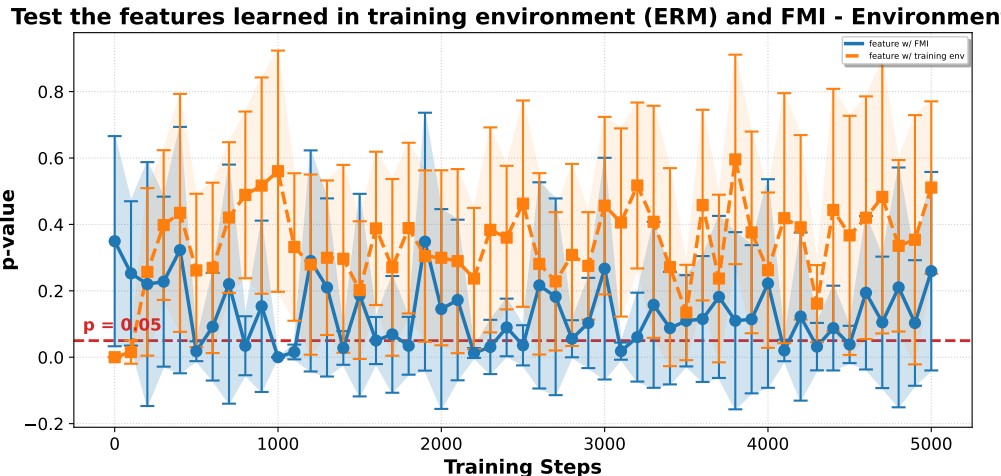

Figure 12: Plots of p-values for testing $Y^e | \hat{f}^e = 1$ in the environment $e = 0.2$ of Colored MNIST given different features. In each plot, the feature learned in the training environment ($e = 0.9$) is colored orange and the feature learned by FMI is colored blue. The y-axis in each plot represents the p-value of the goodness-of-fit test. The dashed red lines represent significance level 0.05. The error bars are obtained by repeating the experiment ten times.

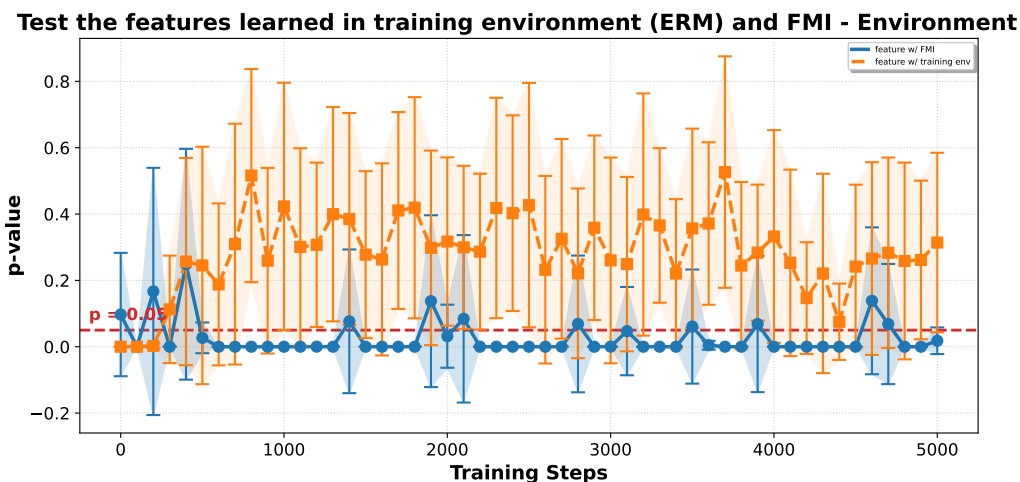

Figure 13: Plots of p-values for testing $Y^e | \hat{f}^e = 1$ in the environment $e = 0.9$ of Colored MNIST given different features. In each plot, the feature learned in the training environment ($e = 0.9$) is colored orange and the feature learned by FMI is colored blue. The y-axis in each plot represents the p-value of the goodness-of-fit test. The dashed red lines represent significance level $0.05$. The error bars are obtained by repeating the experiment ten times.

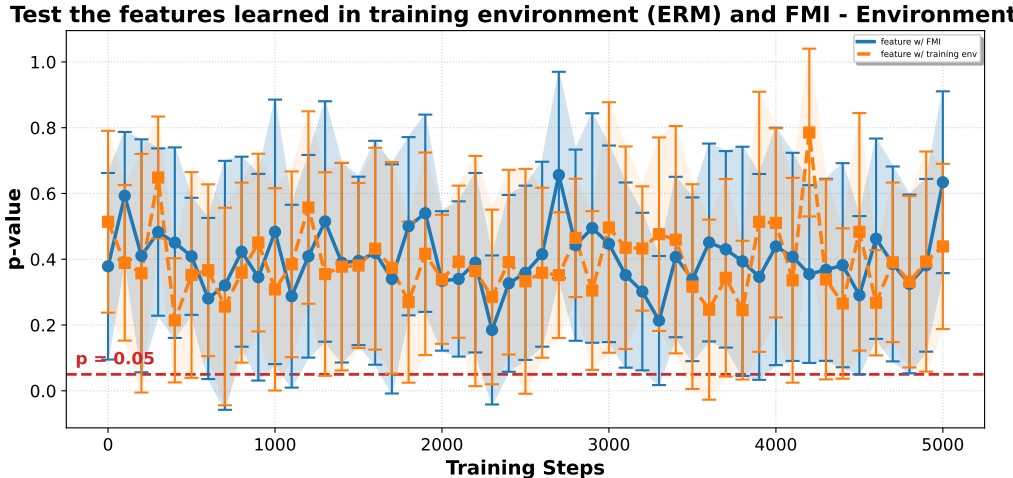

Figure 14: Plots of p-values for testing $Y^e | \hat{f}^e = 1$ in the environment $e = 0.4$ of Colored MNIST given different features. In each plot, the feature learned in the training environment ($e = 0.6$) is colored orange and the feature learned by FMI is colored blue. The y-axis in each plot represents the p-value of the goodness-of-fit test. The dashed red lines represent significance level $0.05$. The error bars are obtained by repeating the experiment ten times.

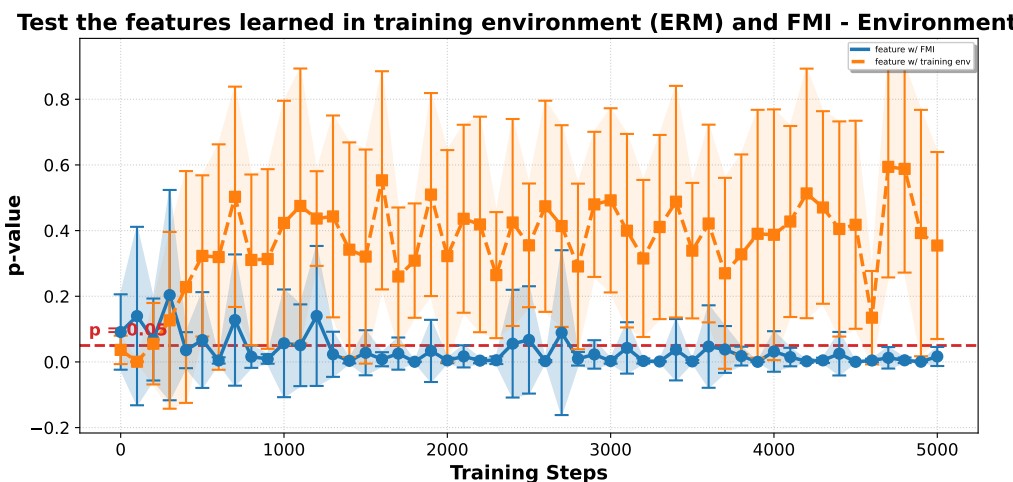

Figure 15: Plots of p-values for testing $Y^e | \hat{f}^e = 1$ in the environment $e = 0.6$ of Colored MNIST given different features. In each plot, the feature learned in the training environment ($e = 0.6$) is colored orange and the feature learned by FMI is colored blue. The y-axis in each plot represents the p-value of the goodness-of-fit test. The dashed red lines represent significance level $0.05$. The error bars are obtained by repeating the experiment ten times.

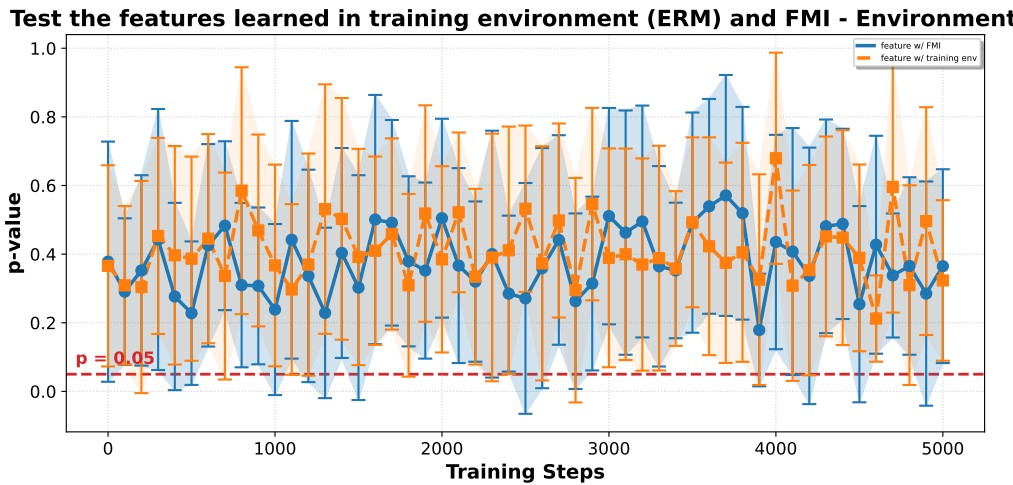

Figure 16: Plots of p-values for testing $Y^e | \hat{f}^e = 1$ in the environment $e = 0.7$ of Colored MNIST given different features. In each plot, the feature learned in the training environment ($e = 0.8$) is colored orange and the feature learned by FMI is colored blue. The y-axis in each plot represents the p-value of the goodness-of-fit test. The dashed red lines represent significance level $0.05$. The error bars are obtained by repeating the experiment ten times.

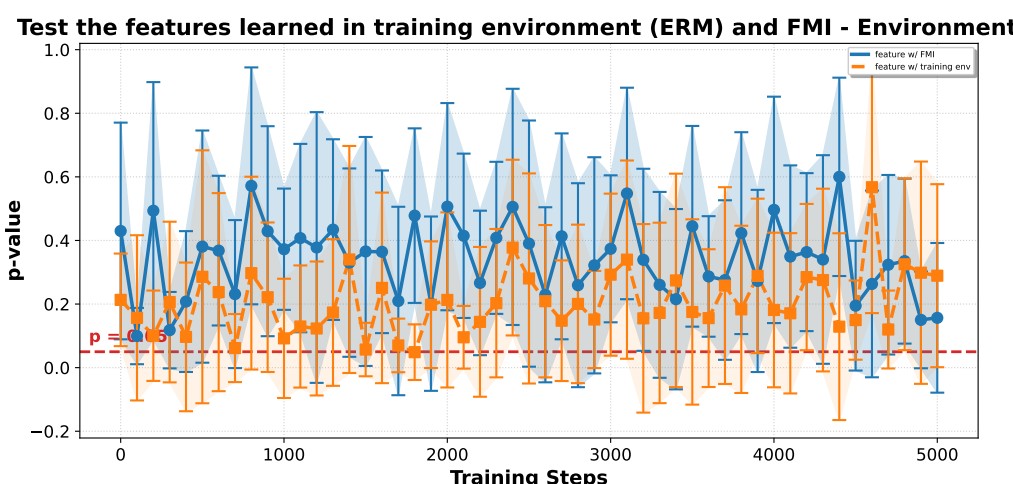

Figure 17: Plots of p-values for testing $Y^e|\hat{f}^e = 1$ in the environment $e = 0.8$ of Colored MNIST given different features. In each plot, the feature learned in the training environment ($e = 0.8$) is colored orange and the feature learned by FMI is colored blue. The y-axis in each plot represents the p-value of the goodness-of-fit test. The dashed red lines represent significance level $0.05$. The error bars are obtained by repeating the experiment ten times.

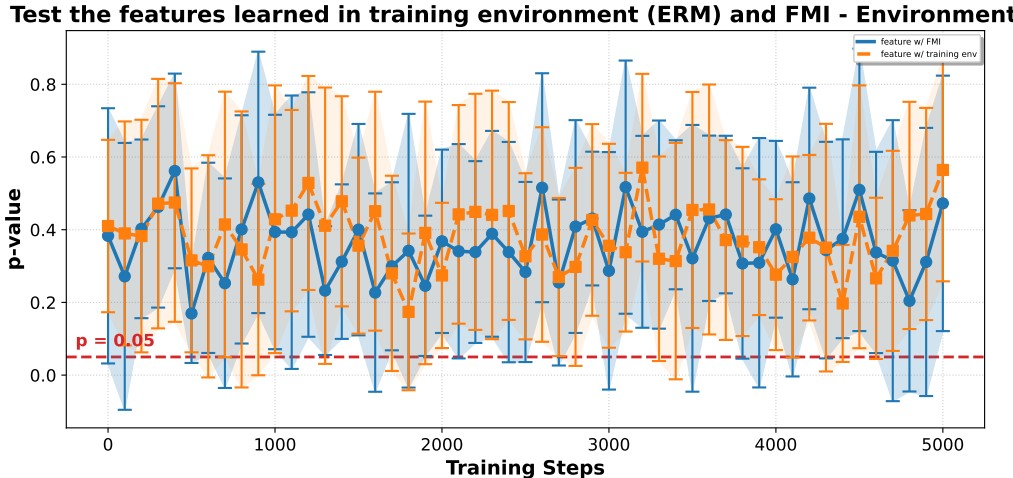

Figure 18: Plots of p-values for testing $Y^e|\hat{f}^e = 0$ in the environment $e = 0.1$ of Colored MNIST given different features. In each plot, the feature learned in the training environment ($e = 0.1$) is colored orange and the feature learned by FMI is colored blue. The y-axis in each plot represents the p-value of the goodness-of-fit test. The dashed red lines represent significance level $0.05$. The error bars are obtained by repeating the experiment ten times.

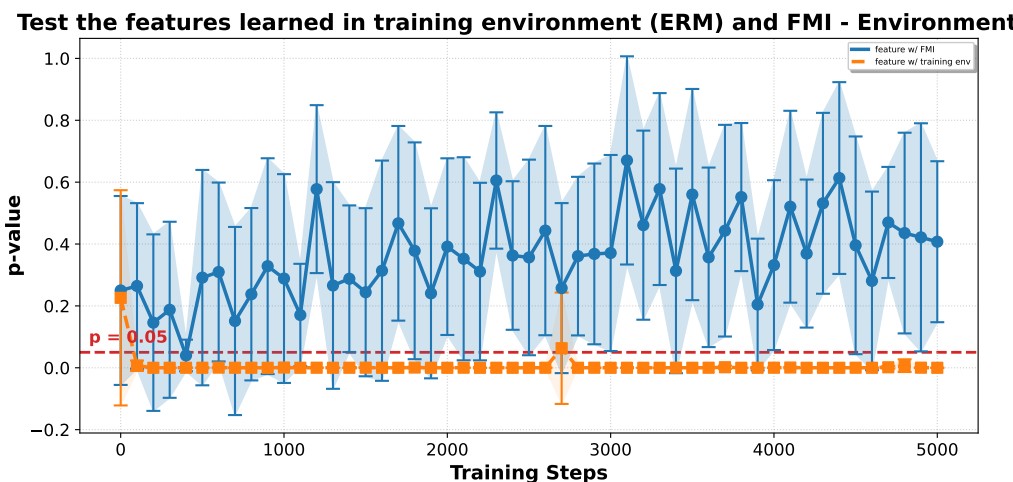

Figure 19: Plots of p-values for testing $Y^e|\hat{f}^e = 0$ in the environment $e = 0.3$ of Colored MNIST given different features. In each plot, the feature learned in the training environment ($e = 0.1$) is colored orange and the feature learned by FMI is colored blue. The y-axis in each plot represents the p-value of the goodness-of-fit test. The dashed red lines represent significance level 0.05. The error bars are obtained by repeating the experiment ten times.

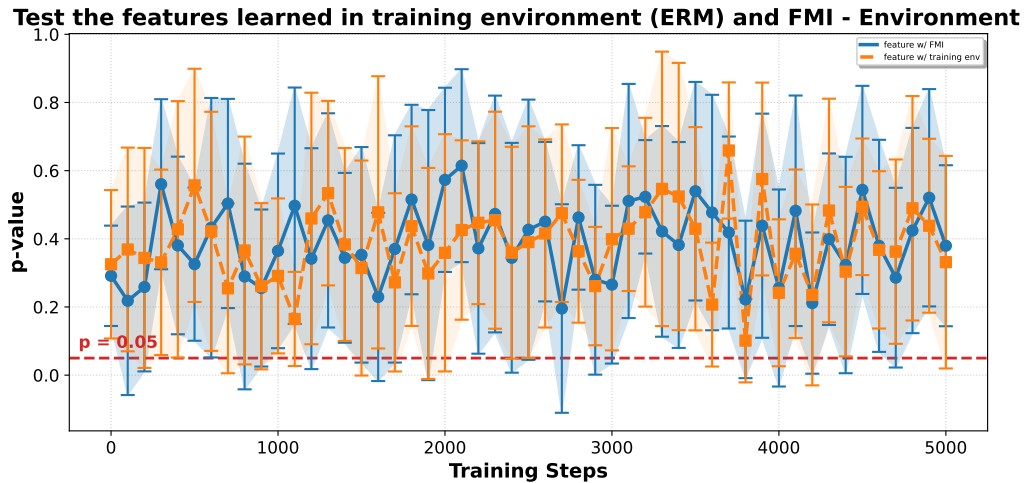

Figure 20: Plots of p-values for testing $Y^e|\hat{f}^e = 1$ in the environment $e = 0.1$ of Colored MNIST given different features. In each plot, the feature learned in the training environment ($e = 0.1$) is colored orange and the feature learned by FMI is colored blue. The y-axis in each plot represents the p-value of the goodness-of-fit test. The dashed red lines represent significance level 0.05. The error bars are obtained by repeating the experiment ten times.

1620
1621
1622
1623
1624
1625
1626
1627
1628
1629
1630
1631
1632
1633
1634
1635
1636
1637
1638
1639
1640
1641
1642
1643
1644
1645
1646
1647
1648
1649
1650
1651
1652

Figure 21: Plots of p-values for testing $Y^e | \hat{f}^e = 1$ in the environment $e = 0.3$ of Colored MNIST given different features. In each plot, the feature learned in the training environment ($e = 0.1$) is colored orange and the feature learned by FMI is colored blue. The y-axis in each plot represents the p-value of the goodness-of-fit test. The dashed red lines represent significance level $0.05$. The error bars are obtained by repeating the experiment ten times.
