# OpenReview forum: "Feature Matching Intervention: Leveraging Observational Data for Causal Representation Learning"
_ICLR.cc/2025/Conference — Submitted to ICLR 2025_

### Official Review · Reviewer_ucNn · 2024-10-28

**Soundness:** 3
**Presentation:** 3
**Contribution:** 3
**Rating:** 6
**Confidence:** 2

**Summary:**

The authors propose Feature Matching Intervention (FMI), which uses a matching procedure to mimic perfect interventions. They define causal latent graphs, extending structural causal models to latent feature space, providing a framework that connects FMI with causal graph learning.

**Strengths:**

The procedure emulates perfect interventions within causal latent graphs. Theoretical results demonstrate that FMI exhibits strong out-of-distribution (OOD) generalizability. Experiments further highlight FMI’s superior performance in effectively identifying causal features solely from observational data.

**Weaknesses:**

Please refer to questions.

**Questions:**

page 3, line 147. ''Thus, identifiability becomes an issue here. However, since our goal is to learn $f\phi$, this concern is not relevant.'' Is the goal to identify $\phi$ here?

---

> ### Author Response · Authors · 2024-11-20
>
> Thank you for your comments!
>
> First of all, we want to emphasize that the primary contribution of this paper lies in causal representation learning, as illustrated in Figure 1 in the manuscript. Causal inference goes beyond prediction. Only causal inference makes it possible to take actions to change the outcome. However, causal relationships cannot generally be identified from observational data alone. Assumptions, such as those involving interventions and the do-operations, are essential for making causal inferences. Rather than assuming a predetermined causal directionality, we leverage the optimal feature learned from the training environment, which corresponds to either the true causal feature ($Z_\text{true}$​) or a spurious feature ($Z_\text{spu}$)​. Additionally, we propose a hypothesis test to evaluate this assumption. Please refer to the new diagram included in the revised manuscript (Figure 2).
>
> Below we provide responses to your question:
>
> **Q1: Is the goal to identify $\phi$ here?**
>
> The goal is not to identify $\phi$. Instead, we aim to learn a better predictor based on $Z_\text{true}$ only.

---

### Official Review · Reviewer_vrs1 · 2024-11-04

**Soundness:** 2
**Presentation:** 2
**Contribution:** 2
**Rating:** 3
**Confidence:** 4

**Summary:**

The paper introduces Feature Matching Intervention (FMI), an approach for mitigating spurious correlations using a feature-matching procedure to mimic perfect interventions on spurious features. The authors provide theoretical guarantees for the proposed method's out-of-distribution (OOD) generalization under specific assumptions and propose a validation approach to assess whether spurious features are being learned in the training environment. Experimental results on synthetic and semi-synthetic datasets, including Colored MNIST and WaterBirds, demonstrate that the proposed method outperforms baseline methods, especially in scenarios with strong spurious correlations in the training data.

**Strengths:**

1. The theoretical analysis of the OOD generalizability of the proposed method is rigorous, and the derivation procedure is clear and easy to follow.

2. The experiments demonstrate that the proposed method outperforms baselines in identifying causal features, especially in the presence of spurious correlations.

**Weaknesses:**

1. __Single Environment Claim__: Although the authors claim that the proposed method can mitigate spurious correlations using data from a single training environment, Assumptions 2 and 3 appear to imply the need for multiple environments when deriving the theoretical guarantees. Additionally, the empirical studies on Colored MNIST utilize two training environments, which seems inconsistent with this claim. It would be beneficial for the authors to conduct experiments using a single training environment and evaluate the method's performance on both synthetic and semi-synthetic datasets.

2. Assumption 1 appears to be more of an intuitive conjecture, lacking formal theoretical support.

3. __Missing Related Work__: Some relevant related works have been omitted. First, the concept of reweighting to mimic perfect interventions on spurious features for improving distributional robustness has been discussed in [1] and [2]. Additionally, there is a body of work focused on improving group distributional robustness based on the understanding that ERM tends to learn spurious correlations ([3], [4], [5]). The proposed method seems to share similarities with these works. It would be helpful if the authors could discuss the novelty of their approach and how it fills a gap compared to these existing works.

4. __Subsampling and Overfitting Concerns__: The authors use subsampling to remove the dependence between the label and spurious features. However, spurious correlations often occur in highly imbalanced data distributions, and subsampling in such cases could lead to dropping a substantial portion of the data from majority groups. This may increase the risk of overfitting, especially if the remaining dataset is small. It would be great if the authors could address how they mitigate the risk of overfitting in this scenario.

5. __Validation Environment Concerns__: When assessing whether spurious features are learned in the training environment, the authors propose using a validation environment. This appears to contradict the single-training-environment assumption. One of the benefits of the single-environment setting is the reduced requirement for environment labels or predefined environment divisions. However, if a validation environment is required, this benefit is lost. Furthermore, the validity of the test may depend on the level of distributional shift between the training and validation environments. If the shift is minimal, the test might incorrectly conclude that ERM has learned the causal feature. Clarification on these points would be great.

6. __Experimental Setup for WaterBirds Dataset__: Could the authors provide more details regarding the experimental setup for the WaterBirds dataset?

7. __Discussion on Poor Performance in Heterogeneous Training Environments__: The experimental results on Colored MNIST indicate that FMI performs poorly when the training environments are highly heterogeneous. Specifically, when training environments are (0.2, 0.9) or (0.1, 0.9) and the test environment is (0.1) or (0.2), the performance degrades. A detailed discussion on the reasons behind this poor performance and potential ways to address it would be helpful.

8. Minor typo: in line 245, $i, j \in \\{1,2\\}$ should be $i, j \in \\{0,1\\}$?



[1] Makar, Maggie, et al. "Causally motivated shortcut removal using auxiliary labels." International Conference on Artificial Intelligence and Statistics. PMLR, 2022.
[2] Veitch, Victor, et al. "Counterfactual invariance to spurious correlations in text classification." Advances in neural information processing systems 34 (2021): 16196-16208.
[3] Liu, Evan Z., et al. "Just train twice: Improving group robustness without training group information." International Conference on Machine Learning. PMLR, 2021.
[4] Kirichenko, Polina, Pavel Izmailov, and Andrew Gordon Wilson. "Last layer re-training is sufficient for robustness to spurious correlations." arXiv preprint arXiv:2204.02937 (2022).
[5] Yang, Yu, et al. "Identifying spurious biases early in training through the lens of simplicity bias." International Conference on Artificial Intelligence and Statistics. PMLR, 2024.

**Questions:**

Please see the questions in Weaknesses.

---

> ### Author Response · Authors · 2024-11-20
>
> Thank you for your comments!
>
> First of all, we want to emphasize that the primary contribution of this paper lies in causal representation learning, as illustrated in Figure 1 in the manuscript. Causal inference goes beyond prediction. Only causal inference makes it possible to take actions to change the outcome. However, causal relationships cannot generally be identified from observational data alone. Assumptions, such as those involving interventions and the do-operations, are essential for making causal inferences. Rather than assuming a predetermined causal directionality, we leverage the optimal feature learned from the training environment, which corresponds to either the true causal feature ($Z_\text{true}$​) or a spurious feature ($Z_\text{spu}$)​. Additionally, we propose a hypothesis test to evaluate this assumption. Please refer to the new diagram included in the revised manuscript (Figure 2).
>
> Below we provide responses to your questions:
>
> **Q1: Single Environment Claim: Although the authors claim that the proposed method can mitigate spurious correlations using data from a single training environment, Assumptions 2 and 3 appear to imply the need for multiple environments when deriving the theoretical guarantees. Additionally, the empirical studies on Colored MNIST utilize two training environments, which seems inconsistent with this claim. It would be beneficial for the authors to conduct experiments using a single training environment and evaluate the method's performance on both synthetic and semi-synthetic datasets.**
>
> When we are training FMI, we mix 2 environments among the 3 environments as training environment. There are 2 reasons why we choose to use this setting:
>
> 1. Most previous methods require multiple training environments. This setting helps improve the performance of many methods (e.g., IRM, IB-IRM) other than FMI.
> 2. When we apply the model selection method (leave-one-domain-out), we need at least three environments.
>
> Notice that we can definitely train FMI on environment 0.1 and test it on environment 0.9. In this case, FMI also outperforms other methods by a significant margin. As shown in Table 7 and Table 8 in the revised manuscript.
>
> Also, Assumption 2 and Assumption 3 are essential only for deriving the theoretical guarantee of FMI. In practice, FMI can be trained as long as we have a single environment. However, it is multiple environments that provide us with the clue for the poor generalizability of the feature learned in the training environment. The assumptions on environments essentially play the same role as the assumptions on (perfect) interventions in some literature about the identifiability of causal representation learning [1][2][3] and thus are inevitable.
>
> [1] Buchholz, Simon, et al. "Learning linear causal representations from interventions under general nonlinear mixing." Advances in Neural Information Processing Systems 36 (2024).
>
> [2] Jiang, Yibo, and Bryon Aragam. "Learning nonparametric latent causal graphs with unknown interventions." Advances in Neural Information Processing Systems 36 (2023): 60468-60513.
>
> [3] Ahuja, Kartik, et al. "Interventional causal representation learning." International conference on machine learning. PMLR, 2023.
>
> **Q2: Assumption 1 appears to be more of an intuitive conjecture, lacking formal theoretical support.**
>
> As indicated in Fig. 1, there are two scenarios for the best feature learned from the training environment: it is either the true feature or a spurious feature. We have provided a hypothesis test to assess this. When we learned part of the true feature from the training environment, running FMI is not necessary. However, if the feature learned from the training environment is spurious, there are no existing methods available to identify causal features, and our proposed FMI offers a solution to fill this gap. Although Assumption 1 is a conjecture in practice, it may be tested with the hypothesis test we suggested in Sec. 5.2. Furthermore, even if Assumption 1 is violated, we can still apply the workflow of FMI, as shown in Fig. 2 in the revised manuscript. If the feature learned in the training environment does not appear to be a spurious one (i.e., it cannot rejected by the hypothesis test), then we can directly use this feature (learned by ERM) and need not conduct FMI.

---

> ### Author Response · Authors · 2024-11-20
>
> **Q3: Missing Related Work: Some relevant related works have been omitted.**
>
> The major advantage of FMI is that it does not require the access to auxiliary label in the training process. All we need is the input variable and the label. Notice that the workflow of FMI is that (as shown in Fig. 2), if the feature learned in the training environment cannot be rejected by the hypothesis test we proposed in Sec. 5.2, then we can use this feature (learned by ERM) in our predictor, since we do not have evidence to believe this feature is spurious based on the training data. On the other hand, if the feature gets rejected, then we can apply FMI to improve this feature.
>
> **Q4: Subsampling and Overfitting Concerns: The authors use subsampling to remove the dependence between the label and spurious features. However, spurious correlations often occur in highly imbalanced data distributions, and subsampling in such cases could lead to dropping a substantial portion of the data from majority groups. This may increase the risk of overfitting, especially if the remaining dataset is small. It would be great if the authors could address how they mitigate the risk of overfitting in this scenario.**
>
> When we were implementing FMI, we adopted the following strategy to address the problem caused by imbalance:
>
> 1. The subsampling procedure is with replacement, so each image could be sampled multiple times
> 2. We set a hyperparameter for the subsampling procedure, which controls the number of subsamples we use to train the neural network in each step (See line 732-733).
>
> **Q5: Validation Environment Concerns: When assessing whether spurious features are learned in the training environment, the authors propose using a validation environment. This appears to contradict the single-training-environment assumption. One of the benefits of the single-environment setting is the reduced requirement for environment labels or predefined environment divisions. However, if a validation environment is required, this benefit is lost. Furthermore, the validity of the test may depend on the level of distributional shift between the training and validation environments. If the shift is minimal, the test might incorrectly conclude that ERM has learned the causal feature. Clarification on these points would be great.**
>
> We don't agree the claim that the requirement of validation environment renders the lost of the benefit of single environment. In fact, we believe it is crucial to include such a validation environment in data collection step. Without validation environment, it is hard to conclude the model learns a "bad" feature from the training data. After all, the domain generalization problem arises from the poor generalizability of the model, which depends on some new environment. Essentially, FMI proposes a pipeline for general domain generalization. More specifically, given a model and an environment different from the training environment, we can first apply the goodness-of-fit test we proposed in this paper. If the model passes the test, then there is no reason to believe the model is bad based on the data at hand. However, if the model cannot pass the test, we can then apply FMI to try to balance the feature learned from the training environment.
>
> We believe the validity of the test may depend on the level of distributional shift and it is an interesting question to check the sensitivity of our test.
>
> **Q6: Could the authors provide more details regarding the experimental setup for the WaterBirds dataset?**
>
> In this experiment, we created two environments based on the background of the images and we used those with water background as training environment and test the accuracy on images with land background. This WaterBirds example highlights the superior performance of our proposed FMI in real-world scenarios, even when assumptions are not fully met. In this experiment, the setting is highly practical: the correlation between the spurious feature and the label is unknown, and the only known factor is that the distribution of the spurious feature (background) differs between training and testing environments. This demonstrates the effectiveness of FMI in achieving superior performance, even when Assumption 1 is violated.

---

> > ### Comment · Reviewer_vrs1 · 2024-11-27
> > **Official Comment by Reviewer vrs1**
> >
> > Thank you for the detailed response. While I appreciate the effort to address my concerns, my primary issue regarding the motivation and significance of the proposed method has not been addressed. Below, I have outlined actionable feedback to clarify and expand on my concerns:
> >
> > 1. __About Assumption 1__:
> > I find Assumption 1 to be quite strong, and I believe it is unrealistic to assume it will hold in real-world scenarios. While the authors mentioned that a validation environment can be collected and statistical testing can be performed to determine whether FMI is necessary, this raises several practical issues:
> > * How would you practically collect a validation environment? For example, in the ColoredMNIST dataset, if the training data is sampled from environment 0.1, then ideally the validation environment should represent environment 0.9. However, in this case, including the validation environment as part of the training data (i.e., training on data sampled from environments 0.1 and 0.9) would allow the use of invariance-based domain generalization methods (e.g., IRM) to achieve invariant representation and mitigate spurious correlations. What advantage does FMI offer over such approaches in this scenario?
> >
> > * If the collected validation environment represents, for instance, environment 0.3 instead of 0.9, the model would more likely pass the statistical test. According to the manuscript, this would suggest that applying ERM is sufficient. However, in such cases, does ERM truly learn a causal representation? It is clear that ERM in this scenario would still rely on spurious correlations (e.g., between color and label) and fail to provide the causal representation, which the manuscript emphasizes as its goal.
> >
> > 2. __Experimental Setup for WaterBirds Dataset__:
> > The response provided regarding the experimental setup for the WaterBirds dataset remains too general, making it difficult to understand the specifics. I have the following questions:
> > * Are you generating the datasets following the procedure described in [1]?
> > * How do you collect the validation environment in this case? Or are you assuming that the training data is inherently highly imbalanced, with spurious correlations between the background and the label?
> >
> > 3. __Related works__:
> > While the proposed method does not rely on auxiliary labels, it is closely related to previous works that aim to mitigate spurious correlations. Specifically, I referenced this line of work in my initial review (point 3 under weaknesses: "Additionally, there is a body of work focused on improving group distributional robustness based on the understanding that ERM tends to learn spurious correlations ([3], [4], [5])."). The manuscript does not sufficiently discuss the relationship between the proposed method and these prior works. It would be helpful if the authors could elaborate on the similarities and distinctions between FMI and this line of research.
> >
> > [1] Sagawa, Shiori, Pang Wei Koh, Tatsunori B. Hashimoto, and Percy Liang. "Distributionally robust neural networks for group shifts: On the importance of regularization for worst-case generalization." arXiv preprint arXiv:1911.08731 (2019).

---

> > > ### Author Response · Authors · 2024-11-27
> > > **Thank you for your response**
> > >
> > > Thank you for your response and clarification on some questions! Here are our replies:
> > >
> > > ## About Assumption 1.
> > >
> > > **Q1: How would you practically collect a validation environment? For example, in the ColoredMNIST dataset, if the training data is sampled from environment 0.1, then ideally the validation environment should represent environment 0.9. However, in this case, including the validation environment as part of the training data (i.e., training on data sampled from environments 0.1 and 0.9) would allow the use of invariance-based domain generalization methods (e.g., IRM) to achieve invariant representation and mitigate spurious correlations. What advantage does FMI offer over such approaches in this scenario?**
> > >
> > > In practice, only a small sample from a validation environment is required. For instance, in both the Colored MNIST and WaterBirds experiments, we calculated the p-values using just 200 images sampled from the validation environment. Including the validation environment as part of the training data does not effectively eliminate the spurious feature, as the sample imbalance issue persists within the training environment. The key advantage of FMI in this scenario is its ability to leverage the small validation sample to extract and utilize information from multiple environments. This capability enables FMI to mitigate spurious features in the training environment more effectively than invariance-based methods.
> > >
> > > **Q2: If the collected validation environment represents, for instance, environment 0.3 instead of 0.9, the model would more likely pass the statistical test. According to the manuscript, this would suggest that applying ERM is sufficient. However, in such cases, does ERM truly learn a causal representation? It is clear that ERM in this scenario would still rely on spurious correlations (e.g., between color and label) and fail to provide the causal representation, which the manuscript emphasizes as its goal.**
> > >
> > > In the scenario where the training environment is $e = 0.1$ and the validation environment is $e = 0.3$, our test is capable of properly rejecting the features learned in the training environment through ERM, as demonstrated in Figures 18--21 of the updated manuscript. However, we acknowledge the possibility of scenarios where the shift between the training and validation environments is too small for the test to reject the ERM-learned features. In such cases, the dataset provides no evidence that the ERM features are problematic. Nonetheless, FMI can still be applied if there is a belief that the learned feature is spurious, enabling further improvement in representation.
> > >
> > > ----
> > > ## Experimental Setup for WaterBirds Dataset
> > >
> > > **Q3: Are you generating the datasets following the procedure described in [1]?**
> > >
> > > Yes, the images were generated following the procedure described in [1]. However, we did not split the dataset into training and testing sets in the same manner. In our experiments, the training images were selected from those with water backgrounds, while the testing images were chosen from those with land backgrounds. This setup provides a scenario where it is uncertain whether the spurious feature (background) is learned. Notably, even in this setting, the performance of FMI remains robust and does not deteriorate compared to other methods, demonstrating its practical applicability.
> > >
> > > **Q4: How do you collect the validation environment in this case? Or are you assuming that the training data is inherently highly imbalanced, with spurious correlations between the background and the label?**
> > >
> > > We do not make any assumptions regarding the training data, as this example is intended solely to demonstrate the effectiveness of FMI in real-world settings. Additionally, there is no validation environment in this particular case.

---

> > > ### Author Response · Authors · 2024-11-27
> > > **Thank you for your response (Cont'd)**
> > >
> > > ## Related Works
> > >
> > > **Q5: Related works: While the proposed method does not rely on auxiliary labels, it is closely related to previous works that aim to mitigate spurious correlations. Specifically, I referenced this line of work in my initial review (point 3 under weaknesses: "Additionally, there is a body of work focused on improving group distributional robustness based on the understanding that ERM tends to learn spurious correlations ([3], [4], [5])."). The manuscript does not sufficiently discuss the relationship between the proposed method and these prior works. It would be helpful if the authors could elaborate on the similarities and distinctions between FMI and this line of research.**
> > >
> > > Thank you for highlighting these related works. We acknowledge that, like the methods in [3], [4], and [5], our approach also involves strategies such as reweighting or resampling to mitigate spurious correlations. However, FMI goes beyond merely improving the prediction accuracy of neural networks. By modeling the data generation process with a causal graph, we demonstrate that FMI actually learns the causal feature under our assumptions. Specifically, the subsampling formula introduced in our paper can be seen as an intervention on the spurious feature, which enables us to break the dependence between the spurious feature and its parents, ultimately leading to the learning of the causal feature. This is the unique contribution of our method.
> > >
> > > Furthermore, upon reviewing [5], we found that the theoretical foundation presented in the paper provides support for the validity of Assumption 1 of FMI. This strengthens our confidence that the assumption holds in practice.

---

> > > > ### Comment · Reviewer_vrs1 · 2024-12-03
> > > > **Official Comment by Reviewer vrs1**
> > > >
> > > > Thank you for the additional response. However, my concerns remain unaddressed.
> > > >
> > > > First, the authors claim that the proposed method learns causal representations. However, when the validation data passes statistical tests, the authors specifically stated:
> > > >
> > > > >__In such cases, the dataset provides no evidence that the ERM features are problematic.__
> > > >
> > > > Based on the response, I do not believe that ERM is guaranteed to learn causal representations in this case. Passing statistical tests on the validation set does not necessarily imply that ERM learns causal representations; it could instead reflect a small distributional shift between the training and validation sets. Moreover, the single-environment assumption seems overly strong, and the effectiveness of the proposed method appears to depend heavily on the quality of the validation set. A more practical scenario assumes that the training data consists of a mixture of multiple environments, where the environment labels are unavailable. The authors should investigate whether the proposed method remains effective under this more realistic assumption and assess its sensitivity to the quality of the validation set.
> > > >
> > > > Second, the authors focus on the proposed method’s ability to mitigate spurious correlations in real-world scenarios. While they acknowledge other approaches that mitigate spurious correlations by reweighting data and retraining models, these methods should be included as baselines. Furthermore, the authors should explicitly explain how their approach differs from these existing methods, rather than simply asserting that it learns causal features. For example: Is the proposed method the only one guaranteed to learn causal features under the assumptions stated in the manuscript? Can any of the baseline methods mentioned also learn causal features under similar assumptions?
> > > >
> > > > Given these unresolved concerns, I maintain my original score.

---

### Official Review · Reviewer_Uddf · 2024-11-04

**Soundness:** 3
**Presentation:** 2
**Contribution:** 3
**Rating:** 5
**Confidence:** 4

**Summary:**

This paper is concerned with representation learning with the aim of finding invariant representations that provide good out of distribution behavior (and can be considered causal under suitable definitions). To achieve this the authors provide a matching scheme which matches on the prognostic score. The authors provide an intuitive and simple realization of the approach, adapting the standard minibatch learning scheme with a subsampling procedure that aims to provide balance and, as a result, control for unobserved confounding. A set of experimental results is provided demonstrating the relative performance of the proposed approach with respect to variants of empirical and invariant risk minimization.

**Strengths:**

* The authors examine an interesting and compelling problem.
* The proposed solution is simple and intuitive; the idea of using matching for this problem holds appeal given both it's relative simplicity and robustness against a broad array of underlying data generating processes.
* Empirical results indicate the proposed approach holds promise.

**Weaknesses:**

While a reader well familiar with this area understands the connections between distribution shift, invariance, and causal inference, it is not made clear within the introduction and problem setting. I would strongly suggest that the authors rewrite these sections making each connection much more explicit. In particular, it should be very explicit what the definition of a causal feature is in this work.

The proposed method, as I understand it is more akin to matching on the prognostic score (Hansen, 2008), rather than more standard matching (e.g., the Stuart paper cited), in that matches are constructed using the _outcomes_ rather than matching covariates with respect to _treatment status_. This should be clarified in the paper. Toward this end, in the problem setup it is stated that these results easily extend to additional outcome types, however it is not immediately clear to me that this should be the case since matching on real valued and multi-valued treatments entails a more nuanced procedure.

Subsampling to make proportions match is reasonable, but also likely introduces issues when there is large distribution skew.

It's not clear to me how equation 5 achieves balance, or why we should think of this as matching in the standard set? Typically we would find matched pairs where $\hat{f}$ is as close as possible, while this doesn't seem to be doing any explicit matching?

Assumption three is incredibly strong, and it is not clear to me how likely this is to hold for any realistic dataset (see below for  a question regarding this). Toward that end, it's not clear to me how substantial the theory is that is provided here. If we are placing strong, and difficult to meet, assumptions on the available data the risk here is that the results serve more as a proof of existence, rather than a general theorem that can be leaned upon in practice.

The highlighting scheme in the results table is confusing. I think the authors meant to bold the best performing method in each setting, rather than just the settings where the algorithm performs well?

Ben B. Hansen, The prognostic analogue of the propensity score, Biometrika, Volume 95, Issue 2, June 2008, Pages 481–488, https://doi.org/10.1093/biomet/asn004

**Questions:**

Can the authors explain how equation 5 is achieving balance here? It's not clear to me that the procedure as described would appropriately control for confounding.

Why is the matching done with respect to batches? It would seem that this would result in poor entailed balance properties?

As I mentioned above Assumption three is incredibly strong, and it is not clear to me how likely this is to hold for any realistic dataset (unless I am misreading it. To be clear, all variables are intervened at all levels? Only one intervention has to be present for each variable? Are they perfect interventions?

Is assumption 4 the observed support?

---

> ### Author Response · Authors · 2024-11-20
>
> Thank you for your comments!
>
> First of all, we want to emphasize that the primary contribution of this paper lies in causal representation learning, as illustrated in Figure 1 in the manuscript. Causal inference goes beyond prediction. Only causal inference makes it possible to take actions to change the outcome. However, causal relationships cannot generally be identified from observational data alone. Assumptions, such as those involving interventions and the do-operations, are essential for making causal inferences. Rather than assuming a predetermined causal directionality, we leverage the optimal feature learned from the training environment, which corresponds to either the true causal feature ($Z_\text{true}$​) or a spurious feature ($Z_\text{spu}$)​. Additionally, we propose a hypothesis test to evaluate this assumption. Please refer to the new diagram included in the revised manuscript (Figure 2).
>
> Below we provide responses to your questions:
>
> **Q1: Can the authors explain how equation 5 is achieving balance here? It's not clear to me that the procedure as described would appropriately control for confounding.**
> Equation (5) gives the subsampling formula for creating an environment $e_m$ by subsampling according to:
> \begin{equation}
>     \begin{aligned}
>         &P^{e_m}(Y = 0|\hat{f} = 0) = \frac{1}{2},\quad  P^{e_m}(Y = 1|\hat{f} = 0) = \frac{1}{2}\\
>         &P^{e_m}(Y = 0|\hat{f} = 1) = \frac{1}{2},\quad P^{e_m}(Y = 1|\hat{f} = 1) = \frac{1}{2}.
>     \end{aligned}
> \end{equation}
> Here, $\hat{f}$ denotes the ERM solution based on the training environment. If $\hat{f}$ learns the spurious features, then Equation (5) ensures that the label $Y$ is independent of the ERM solution $\hat{f}$. Since $\hat{f}$ is based on spurious features, this effectively balances the spurious features.
>
> **Q2: Why is the matching done with respect to batches? It would seem that this would result in poor entailed balance properties?**
>
> The major advantage of matching with respect to batches is that we can train two networks together in the training process. Also, to make the subsample balanced, we mentioned in the Appendix (lines 732-733), "For FMI, we chose a batch size of 64 and conducted subsampling each time we collected at least 32 inputs in each predicted group. "Although this setting is specific to ColoredMNIST, the hyperparameter (group size) can be adjusted. For datasets with many classes, we can set the group size to be moderately large to ensure there are enough samples in each class during the subsampling process.
>
> **Q3: As I mentioned above Assumption three is incredibly strong, and it is not clear to me how likely this is to hold for any realistic dataset (unless I am misreading it. To be clear, all variables are intervened at all levels? Only one intervention has to be present for each variable? Are they perfect interventions?**
>
> We want to emphasize that such assumption is necessary for deriving causality. The assumptions on environments essentially play the same role as the assumptions on (perfect) interventions in some literature about the identifiability of causal representation learning [1][2][3] and thus are inevitable. Notice that Assumption 3 only requires the overall set of environment (through intervention) and we do not assume specific type of intervention in the training environment.  Furthermore, Assumption 3 is essential only for deriving the theoretical guarantee of FMI. In practice, FMI can be trained as long as we have a single environment. However, it is multiple environments that provide us with the clue for the poor generalizability of the feature learned in the training environment (based on the validation environment, we may test the validity of the feature learned in the training environment).
>
> [1] Buchholz, Simon, et al. "Learning linear causal representations from interventions under general nonlinear mixing." Advances in Neural Information Processing Systems 36 (2024).
>
> [2] Jiang, Yibo, and Bryon Aragam. "Learning nonparametric latent causal graphs with unknown interventions." Advances in Neural Information Processing Systems 36 (2023): 60468-60513.
>
> [3] Ahuja, Kartik, et al. "Interventional causal representation learning." International conference on machine learning. PMLR, 2023.
>
> **Q4: Is assumption 4 the observed support?**
>
> The support in Assumption 4 is the support of random variables. Our theoretical guarantee is for infinite sample and therefore this assumption is not with respect the observation.

---

### Official Review · Reviewer_qLzk · 2024-11-06

**Soundness:** 3
**Presentation:** 3
**Contribution:** 3
**Rating:** 3
**Confidence:** 4

**Summary:**

This paper studies settings where images are created from spurious and true features, the true features being invariant across environments. Using a single environment, and under the assumption that _only_ the spurious feature is used for minimising the risk, the authors propose a scheme to create a new dataset (or batch) that simulates interventions on the spurious features. Another model can be trained on this dataset (batch) that then is independent of the spurious feature and only uses the true feature for the task at hand. The authors also introduce a test for the assumption that only the spurious features are used for the task.

**Strengths:**

Overall the idea is a simple one and quite interesting. I do have some issues with the experiments and the test for the assumptions. I think these points could be a lot stronger and should clearly show when the FMI method works and when it doesn't.

- The method is simple and sound when the assumptions of the method hold. The assumptions are _mostly_ clear.
- The paper is mostly clear, although certain areas could be improved (see below)

**Weaknesses:**

The main weakness of the work is that it only works if the model only uses the spurious feature in the training environment. This is quite a strong assumption and thus should be main and centre in the work. In my opinion, in most cases, it seems likely that a model trained on a single environment in this setting will learn from a _mixture_ of spurious and true features (with varying strengths). In this case, applying FMI can also _hurt_ performance as the signal from the true feature can be lost in the matching process. Furthermore, I'm not sure if the test in Section 5.2 will pick up this case, as Y|Z may differ in the validation environment even in the case that both spurious and true features are used. A thorough analysis of this case will greatly improve the work. It would be of interest to see how sensitive the test is and how much performance is lost if the training results in a mixture of true and spurious features. I would encourage the authors to discuss if this is the case, and include experiments that show if performance drops or not (for example when colour noise in Section 6.2 is higher than the label noise), and to show how trustworthy their proposed test is at finding these cases.

A second weakness is that the procedure requires training a neural network to convergence at every training step.

**Questions:**

- What is the resultant added cost in your experiments as you require training a network to convergence at every step?
- Is it not possible to just train two neural networks to convergence instead of training a new one to convergence at every training step?
- L312: I'm not sure how Assumption 3 implies that Zspu is the feature learned in the training environment? Surely this depends on how correlated the spurious feature is with the label in the training environment? As far as I can tell, there is no assumption about the training environment at all.
- L345: Related to the first weakness: This property may still hold if _both_ the spurious feature and the true feature are used. I think it may be more correct to say that if Y|Ze and Y|Ze0 are the same then you can be sure that Z is the true feature.
- The experiments in 6.1 an 6.3 are not very informative. There is no clear information from what I can see about the level of correlations between the spurious features and labels.
- Section 6.2: The Colored MNIST setting does not read clearly to me at all. I have a few questions about this:
- There are 3 environments (0.1, 0.2, 0.9) it sounds like two are used as the training environment and one is used as a testing environment. Are the two that are used as a training environment mixed? If so, why is this done? Why not train on 0.1 and then test on 0.9 and so on? This is not commented on at all. This seems like an odd choice and makes the experiment quite unclear.
- It seems that the 0.1 and 0.9 setting are the same (as they have the same correlation between label and spurious feature), is this correct?
- The performance drop when 0.1 is the test environment is a bit worrying. I completely see why training on (mixture of) 0.1 and 0.2 would result in the spurious feature being used, and the performance on the 0.9 environment improves when FMI is used. I'm not sure I believe the claim in L418 that the performance drop in the 0.1 env is due to subsampling. It seems that an equally reasonable explanation could be that training on 0.9 and 0.2 together results in a classifier that uses _both_ Ztrue and Zspu. Matching would thus result in a drop in performance. This should be tested thoroughly to see if this is the case, and to see if the test (Section 5.2) actually spots when this is the case.
- The plot in figure 5 is unclear to me. What is environment 0 and environment 1? Why are there two plots given that you are testing how similar Y|f are in two different environments?

---

> ### Author Response · Authors · 2024-11-20
>
> Thank you for your comments!
>
> First of all, we want to emphasize that the primary contribution of this paper lies in causal representation learning, as illustrated in Figure 1 in the manuscript. Causal inference goes beyond prediction. Only causal inference makes it possible to take actions to change the outcome. However, causal relationships cannot generally be identified from observational data alone. Assumptions, such as those involving interventions and the do-operations, are essential for making causal inferences. Rather than assuming a predetermined causal directionality, we leverage the optimal feature learned from the training environment, which corresponds to either the true causal feature ($Z_\text{true}$​) or a spurious feature ($Z_\text{spu}$)​. Additionally, we propose a hypothesis test to evaluate this assumption. Please refer to the new diagram included in the revised manuscript (Figure 2).
>
> Below we provide responses to your questions:
>
> **Q1: What is the resultant added cost in your experiments as you require training a network to convergence at every step? Is it not possible to just train two neural networks to convergence instead of training a new one to convergence at every training step?**
>
> In practice, we do not require training a network to convergence at every step. As shown in Appendix A.3, we tried three different training strategies:
>
> 1. Train subnetwork and the main network together for 5,000 steps. In each step, we update both subnetwork and main network and use the classification result of the subnetwork to conduct subsampling;
> 2. Train subnetwork for 4,000 steps to warm up. Then we use the classification result of the subnetwork to conduct subsampling and train the main network for 4,000 steps;
> 3. Train subnetwork for 5,000 steps to warm up. Then we use the classification result of the subnetwork to conduct subsampling and train the main network for 5,000 steps;
>
> The experiment results we reported in the main text is based on strategy 1 and in this case, we train the two networks together and does not require the main network to converge in each step. Similarly, if we apply strategy 2 or 3, we still need not add this convergence requirement. The additional costs in FMI are two-folded: (1) the cost of training two neural networks; (2) the cost of subsampling. However, we can show in our experiments that when the number of classes is small, this additional cost is not significant. For example, in the WaterBirds experiment, we did not observe a substantial increase in cost compared to ERM. Specifically, the running time of FMI over 5 repetitions ranges from approximately 40 to 70 minutes, while the running time of ERM over 5 repetitions is around 50 minutes. Additionally, FMI is significantly faster than some previous methods. For instance, the running time of IGA over 5 repetitions is approximately 140 minutes.
>
> **Q2: I'm not sure how Assumption 3 implies that $Z_\text{spu}$ is the feature learned in the training environment? Surely this depends on how correlated the spurious feature is with the label in the training environment? As far as I can tell, there is no assumption about the training environment at all.**
>
> We agree that Assumption 3 does not imply that $Z_\textup{spu}$ is the feature learned in the training environment. The role of $Z_\textup{spu}$ was given in Assumption 1. Additionally, Assumption 1 implicitly requires that in the training environment, $Z_\textup{spu}$ must be strongly correlated with the label. As indicated in Fig. 1, there are two scenarios for the best feature learned from the training environment: it is either the true feature or a spurious feature. We provided a hypothesis tests to assess this.
>
> **Q3: Related to the first weakness: This property may still hold if both the spurious feature and the true feature are used. I think it may be more correct to say that if $Y|Z^e$ and $Y|Z^{e_0}$ are the same then you can be sure that Z is the true feature.**
>
> Your suggested statement is also correct. Our statement, as given in Proposition 1, is 'If $Z^{e_0}$ is spurious feature, then there must exist a corresponding validation environment', which is the contrapositive of your statement and therefore is equivalent to your statement. However, it is worth noting that the following statement: 'if there exists a validation environment, then the feature learned in the training environment is the spurious feature', is incorrect.

---

> ### Author Response · Authors · 2024-11-20
>
> **Q4: The experiments in 6.1 and 6.3 are not very informative. There is no clear information from what I can see about the level of correlations between the spurious features and labels.**
>
> For experiment 6.1, the correlation between the spurious feature and label comes from an anti-causal setting, where the generated data can be well predicted by the spurious feature. For experiment 6.2, we created two environments based on the background of the images and we used those with water background as training environment and test the accuracy on images with land background. The setting of this experiment is more practical, where the correlation between the spurious feature and label is unknown and the only thing we know is that the distribution of the spurious feature (background) varies across training and testing environment. This helps us understand the effectiveness of FMI even when Assumption 1 is violated.
> The details of experiments 6.1 and 6.3 can be found in Appendix A. The correlation between the spurious feature and label can be explicitly derived based on line 660- 676 for experiment 6.1.
>
> **Q5: There are 3 environments (0.1, 0.2, 0.9) it sounds like two are used as the training environment and one is used as a testing environment. Are the two that are used as a training environment mixed? If so, why is this done? Why not train on 0.1 and then test on 0.9 and so on? This is not commented on at all. This seems like an odd choice and makes the experiment quite unclear.**
>
> When we are training FMI, we mix 2 environments among the 3 environments as training environment. There are 2 reasons why we choose to use this setting:
>
> 1. Most previous methods require multiple training environments. This setting helps improve the performance of many methods (e.g., IRM, IB-IRM) other than FMI.
> 2. When we apply the model selection method (leave-one-domain-out), we need at least three environments.
>
> Notice that we can definitely train FMI on environment 0.1 and test it on environment 0.9. In this case, FMI also outperforms other methods by a significant margin. As shown in Table 7 and Table 8 in our revised manuscript.
>
> **Q6: It seems that the 0.1 and 0.9 setting are the same (as they have the same correlation between label and spurious feature), is this correct?**
>
> In terms of the strength of the correlation between color and label, they are the same. However, in $e = 0.1$ and $e = 0.9$, the color correlated with $Y = 1$ is different (green in $e = 0.1$ and red in $e = 0.9$). Therefore, if we train the network with ERM in $e = 0.1$ and test it in $e = 0.9$, the model that uses color to make prediction would get predictions complement to the true label. This can be verified by the results shown in Table 7. ERM trained in environment $e = 0.1$ has prediction accuracy around $10\%$ in environment $e = 0.9$, which is about the same as the proportion of the green images in group $Y = 1$.
>
> **Q7: The performance drop when 0.1 is the test environment is a bit worrying. I completely see why training on (mixture of) 0.1 and 0.2 would result in the spurious feature being used, and the performance on the 0.9 environment improves when FMI is used. I'm not sure I believe the claim in L418 that the performance drop in the 0.1 env is due to subsampling. It seems that an equally reasonable explanation could be that training on 0.9 and 0.2 together results in a classifier that uses both $Z_\text{true}$ and $Z_\text{spu}$. Matching would thus result in a drop in performance. This should be tested thoroughly to see if this is the case, and to see if the test (Section 5.2) actually spots when this is the case.**
>
> As pointed out in the question, when training on mixture of $e = 0.2$ and $e = 0.9$, we can conduct the hypothesis test for ERM feature and FMI feature (as our first step in the workflow). Not surprisingly, the feature learned by ERM in this case cannot be rejected in testing environment. As a result, the workflow (Figure 2 in the manuscript) suggests us use ERM feature directly. The test results can be found in Figure 11, Figure 12 and Figure 13 of the revised manuscript.
>
> **Q8: The plot in figure 5 is unclear to me. What is environment 0 and environment 1? Why are there two plots given that you are testing how similar $Y|f$ are in two different environments?**
> For Figure 5, we first train the model in training environment $e = 0.1$ and use $e = 0.9$ as testing environment. The left plot in Figure 5 represents the p-value of the goodness-of-fit test in training environment (denoted by $0$) and the right plot represents the p-value of the goodness-of-fit test in testing environment (denoted by $1$). Also, the blue line represents the model trained by FMI and the orange line represents the model trained by ERM. The test that really matters is the right one, as it uses two different environments. The first one was included simply as a comparison.

---

> > ### Comment · Reviewer_qLzk · 2024-11-24
> > **Response to Authors**
> >
> > > Q1
> >
> > My confusion stems from Line 5 in Algorithm 1 which states that parameters of $f_1$ should update until convergence. If the networks are trained jointly, the authors might consider changing this line to be less vague.
> >
> > > Q5
> >
> > Yes this makes sense, although the choices of the numbers 0.1, 0.2, 0.9 are less clear to me. It might make more sense to have a  *separate experiment* testing FMI vs. ERM where they are clearly trained on 1 environment and tested on another. Any other baseline that only requires one environment might also be included here. The key here is to show that FMI can actually outperform ERM with a single environment.
> >
> > > Q8
> >
> > I see what this figure is saying, however this is not clear from the text or the figure caption at all. The caption should state what the blue and orange lines are, the exact environment used, what env 0 and env 1 are. It's not clear at all that env 0 is the training environment here. I would also suggest writing down in the caption what the reader should learn from the figure.
> >
> > The rest of the comments are clearer.
> >
> > I think my *main concern* is that Assumption 1 is very strong. The main reason for this is that the features extractor may learn some feature (in some task where additive features are optimal), $\phi(Z) = \alpha Z_{true} + \beta Z_{spu}$. Assumption 1 implies that $\alpha = 0$ in the training environment. However, I think it is much more likely that it learns some mixture of the two $Z_{true}, Z_{spu}$, possibly where $\Z_{spu}$ is weighted more strongly than $Z_{true}$. My concern then is that applying FMI to this will hurt performance. In these cases, the question that still remains is then how the test will behave in these cases.
> >
> > I like the workflow that the authors have introduced in Figure 2 in the updated manuscript. However, the question now remains for me is how sensitive is the test such that it will point to the optimal algorithm (FMI. vs. ERM). For example, the authors might consider trying different test and train environments, computing the score for FMI and ERM and stating whether their workflow suggests they should use FMI or ERM. For example in the colored mnist example, if you train on 0.4 and test on 0.6, what is the optimal algorithm, and what does the workflow point to? What about train 0.7 and test 0.8?

---

> > > ### Author Response · Authors · 2024-11-25
> > > **Thank you for your response**
> > >
> > > Thank you for your response and clarification on some questions! Here are our replies:
> > >
> > > **Q1**
> > > We have revised Algorithm 1 to make it clearer that the two networks can be trained jointly.
> > >
> > > ---
> > >
> > > **Q5**
> > > In our updated manuscript, Tables 7 and 8 present the results of all methods trained in a single environment and tested in another. FMI outperforms other methods by a significant margin.
> > >
> > > ---
> > >
> > > **Q8**
> > > We have updated the caption of Figure 5 in our manuscript.
> > >
> > > ---
> > >
> > > We conducted two additional experiments to show our workflow:
> > >
> > > **FMI workflow when training environment is 0.6 and testing environment is 0.4**
> > >
> > > In this case, the feature learned through ERM cannot be rejected and therefore we recommend to use ERM directly. The test results can be found in Figure 14 and Figure 15 in the updated manuscript.
> > >
> > > **FMI workflow when training environment is 0.8 and testing environment is 0.7**
> > >
> > > In this case, the feature learned through ERM cannot be rejected and therefore we recommend to use ERM directly. The test results can be found in Figure 16 and Figure 17 in the updated manuscript.

---

> > > ### Author Response · Authors · 2024-11-27
> > > **Regarding Assumption 1**
> > >
> > > One of the references [1] mentioned by Reviewer 1 provides some theories that support our Assumption 1, where the authors proved that neural networks trained through ERM are more likely to learn the spurious features in practice.
> > >
> > > [1] Yang, Yu, et al. "Identifying spurious biases early in training through the lens of simplicity bias." International Conference on Artificial Intelligence and Statistics. PMLR, 2024.

---

### Meta-Review · Area_Chair_NL8V · 2024-12-21

**Metareview:**

This paper introduces Feature Matching Intervention (FMI), an approach for mitigating spurious correlations using a feature-matching procedure that aims to find invariant representations that provide good out of distribution generalization. Only one reviewer is weakly in favour of acceptance, and did not argue for acceptance.

**Additional Comments On Reviewer Discussion:**

There was some discussion that did not ultimately sway either reviewer.

---

### Decision · Program_Chairs · 2025-01-22

Reject